# Estrogen Receptor-α Targeting: PROTACs, SNIPERs, Peptide-PROTACs, Antibody Conjugated PROTACs and SNIPERs

**DOI:** 10.3390/pharmaceutics14112523

**Published:** 2022-11-19

**Authors:** Arvind Negi, Kavindra Kumar Kesari, Anne Sophie Voisin-Chiret

**Affiliations:** 1Department of Bioproduct and Biosystems, Aalto University, 00076 Espoo, Finland; 2Department of Applied Physics, School of Science, Aalto University, 02150 Espoo, Finland; 3CERMN (Centre d’Etudes et de Recherche sur le Médicament de Normandie), Normandie University UNICAEN, 14000 Caen, France

**Keywords:** peptide PROTACs, SNIPERs, PORTACs, antibody-based PROTAC conjugate, clinical PROTACs

## Abstract

Targeting selective estrogen subtype receptors through typical medicinal chemistry approaches is based on occupancy-driven pharmacology. In occupancy-driven pharmacology, molecules are developed in order to inhibit the protein of interest (POI), and their popularity is based on their virtue of faster kinetics. However, such approaches have intrinsic flaws, such as pico-to-nanomolar range binding affinity and continuous dosage after a time interval for sustained inhibition of POI. These shortcomings were addressed by event-driven pharmacology-based approaches, which degrade the POI rather than inhibit it. One such example is PROTACs (Proteolysis targeting chimeras), which has become one of the highly successful strategies of event-driven pharmacology (pharmacology that does the degradation of POI and diminishes its functions). The selective targeting of estrogen receptor subtypes is always challenging for chemical biologists and medicinal chemists. Specifically, estrogen receptor α (ER-α) is expressed in nearly 70% of breast cancer and commonly overexpressed in ovarian, prostate, colon, and endometrial cancer. Therefore, conventional hormonal therapies are most prescribed to patients with ER + cancers. However, on prolonged use, resistance commonly developed against these therapies, which led to selective estrogen receptor degrader (SERD) becoming the first-line drug for metastatic ER + breast cancer. The SERD success shows that removing cellular ER-α is a promising approach to overcoming endocrine resistance. Depending on the mechanism of degradation of ER-α, various types of strategies of developed.

## 1. Introduction

Estrogen receptors (ER) belong to the nuclear superfamily with ligand-dependent functioning. Estrogen receptors are mainly present in the nucleus and on the plasma membrane or the phospholipids’ membrane of the cell. The normal homeostasis signaling of estrogen receptors plays an essential role in growth, development, differentiation, and regulatory functions associated with the reproductive systems of both females and males. In 1962, Jenson and Jacobson demonstrated that estradiol binds to the estrogen receptor (ER) even in the uterus, vagina, and pituitary gland [1]. Any aberration in the signaling of estrogen receptors leads to various types of endocrine disorders and associated cancers, for example, breast cancer, endometrial cancer, and osteoporosis, where the estrogen receptor is a designated clinical target [2]. Most notably, 70% of breast cancers have over-expressed estrogen receptors (ER+), which display its significance as a therapeutic target in breast cancer. Estrogens show these activities by binding with the ER that functions as signal transducers and transcription factors in order to modulate the expression of target genes [3].

## 2. Structure of Estrogen Receptors: Specificity of Ligand Binding Domain

ER constitute 12 helices; when an agonist is bound with ER-α in a three-layered structure, helices 4, 5, 6, 8, and 9 are present at one layer, whereas H1 and H3 are present on one side. The ERs are mainly classified into three classes (a) ER-α, (b) ER-β (c) ER-γ. ER-α and ER-β are encoded from ESR1 and ESR2 genes [4]. However, there is no clear evidence for the genes of ER-γ. The first ER was identified in 1958 by Elwood Jensen and colleagues, named ER-α. ER is composed of six domains that provide specific functional roles, those being (a) N-terminal domain (NTD), (b) DNA-binding domain (DBD), (c) ligand binding domain (LBD), (d) The hinge region, (e) F-domain or C-terminal domain. The six domains are shown in Figure 1. Ligand-independent activation function (AF-1) encoded by the N-terminal domain [5] provides a region for protein-protein interaction [6]. The “D” domain, also known as the hinge region, which follows DBD, contains nuclear localization signal. It is a flexible region between DBD and LBD, with 36% homology shared between ER-α and ER-β. A sequence of 40–50 amino acid residues in the D-domain separates the LBD and DBD, which is necessary for receptor dimerization. C- terminal domain or F-domain contains 42 amino acids, followed by LBD. The LBD identified various compounds dissimilar in shape, dimension, conformational and chemical properties. There are two activation domains, AF-1 and AF-2 located in between NTD and LBD, respectively, that regulate the transcription of ER. A full transcription of ER is due to synergism between the two AFs (AF-1 and AF-2 domain) AF-1 domain is hormone-independent, whereas the AF-2 domain is hormone-dependent. DBD contains two zinc finger motifs that directly interact with the DNA helix. The F-domain plays an important role in differentiating the ligands, whether it is an estrogen agonist or antagonist. Ligands such as estradiol or diethylstilbestrol (DES) act as pure agonists, whereas fulvestrant acts as a pure antagonist. SRC-1 (steroid receptor coactivator-1)/N-CoA 1 [7], GRIP 1/TIF2/N-CoA 2 attach with ER-α in a ligand-dependent manner. These proteins behave as co-activators because they stimulate the transcription of ER-α and other NRs. The amino acid sequence in the H3 helix is Val-355, Leu-354, Ile-358, Lys-362, and Ala-361, whereas sequence in H5 helix is Glu-375, Val-376, Glu-380 and Leu-379. Leu-372 is present in between H3 and H4 helix of ER-α. In H12, amino chain is Met-543, Glu-542, Leu-539 and Asp-538. H12 is necessary for conformational change due to which dimerization takes place. All these helices are present in binding site of NR box II peptide which is similar to a shallow groove. 

The sides and lower surfaces of the groove are entirely hydrophobic except for the terminations attributed to their charged character. The LBD interacts with Leu-690, Leu-693, Leu-694, and Ile-689 of the NR box II peptide α by Vander Waals force of attraction. Compared to ER-β, which was a second estrogen receptor discovered in rat prostate named ER-β in 1996. Unexpectedly, ER-β was shown opposite effects to ER-α. Both receptors are distributed unevenly in the organs, such as ER-α encoded by different genes, and are located on different chromosomes, tissues, and organs. ER-α is found in the endometrium, breast, ovary, hypothalamus, uterus, skin, guts, and ER-β in ovarian granulosa cells, kidney, brain, bone, heart, lungs [8], intestinal mucosa, prostate, endothelial cells, adrenal, skin, pituitary gland. In ER-α, AF-1 domain is very active in the initiation of receptor-gene expression, whereas in the case of ER-β, this domain is almost inactive under the same conditions. ER-α and ER-β both receptors are similar in size; they contain approximately 600 and 530 amino acids, respectively. The DNA binding domain (DBD) possesses a two zinc-finger structure that plays a significant role in receptors’ dimerization process and receptor binding with ERE elements and specific DNA sequences [9]. ER-α act as a transcriptional activator on ERE, which is a palindromic sequence made up of two hexanucleotide sequence, whereas the ER-β acts as a suppressor, so that the function of ER-α is suppressed by dimerization with ER-β. When a ligand binds with the estrogen receptor site, heat shock protein (such as hsp90) will be dissociated from the receptor and then receptor will be activated. After receptor activation, there is a change in conformation of the site of estrogen receptor due to which both receptors are bind with each other thus; we can say dimerization process takes place. After dimerization, it binds with ERE element or on the DNA sequence, transcription phenomenon occurs. The region between 91- 121 amino acids is required to generate the most significant transcription activity, and the region between the 41–150 amino acids is required for the AF-1 domain. 

## 3. Signaling of ER

The mechanism of action of ER is of the following different types, as shown in Figure 2:(1)Classical ligand-dependent signaling(2)DNA-binding independent signaling(3)Ligand—independent signaling(4)Non-genomic signaling

### 3.1. Classical Ligand-Dependent

If the ligand is hydrophobic, then ligand crosses the cell membrane and binds with ER in the nucleus. During binding, the heat shock protein (such as hsp 90, hsp70) will be dissociated from ER, and the receptor will be activated. Activated ER tries to bind with another ER present in the nucleus because of the conformational change in the receptor site then, finally, it will be bind with each other. After binding, it binds to the ERE elements present on the DNA sequence. The DNA –bound receptors affect the transcription via direct or indirect co-factor proteins [10] SRC-1, GRIP-1, TRAP 220, CBP/p300, and p68 RNA helicase [11]. The interaction between ER and co-activators stabilizes the pre-initiation step of transcription and smooth the progress of distraction of chromatin at the ERE. The ERKO mice model is helpful in confirming the ligand-dependent ER signaling through the ERE-mediated mechanism in in vitro studies.

### 3.2. Ligand Independent

In the absence of a ligand, ER can be activated by extracellular signals through phosphorylation. Extracellular signals involve the EGFR, cell cycle regulators, IGF-1 [12], peptide growth factors, cytokines, neurotransmitters [13,14], and some analogs of intracellular signals such as 8-bromo-cyclic-adenosine monophosphate that helps to activate the ER [15]. Protein kinase C (PKC) [16] and protein kinase A (PKA) [17] are the regulators of cellular phosphorylation that play a significant role in ER activation. These signals increase the target gene response of ER. The activation of ER via growth factor requires AF-1 domain containing N-terminal, whereas intracellular signaling through cAMP requires AF-2 domain of the receptor. Within the A/B domain, Cyclin A2, CDK-2, Ser-118, Ser-104, Ser-106, Ser-154, Ser-167 residues are essential targets for the phosphorylation of MAPK pathway even in the presence of E_2_, succeeding concern with EGF and IGF. After this, the receptor will interact with p68 co-activator with RNA helicase and activated the transcription. MAPK pathway recruits SRC-1 to N-terminal and stimulates the ER-β murine activity. Co-activators play a significant role in transcription through the stimulation of ER and other signaling pathways such as kinase pathway, and growth factors pathway [18]. 

### 3.3. ERE-Independent Genomic Signaling of ER

This mechanism activates ER-α through IGF-1 and collagenase expression via receptor interaction with Jun and Fos at AP-1 binding sites. Various genes having GC-rich promoter series are triggered via ER-α-Sp1 complex. Both binding domains such as AF-1 and AF-2 of the receptor are required for the AP-1 responsive element of E_2_- ER-α activation that binds and stimulates the activity of SRC-1 and GRIP-1 recruited by Fos/Jun. Interestingly, if the AF-1 domain is absent in ER-β, that is not able to initiate the transcription of AP-1 regulated genes. If an agonist is bound with ER, there is a chance of performing different physiological actions through the regulation of a distinctive gene subset [19]. The interaction between ER and AP-1 pathways has been reported to be more complicated in in-vitro conditions. ERKO model is an irreplaceable mechanism that contributes to estrogen signaling.

### 3.4. Non-Genomic Signaling

In the breast and nervous system, ER shows non-genomic signaling through the plasma membrane ER that is associated with intracellular signal transferring genetic material from one part to another related proteins. It is closely associated with the various kinase activations such as MAPK, protein kinases [20], Akt. It is mediated by classical ERs (ER-α and ER-β) that are present on the cell membrane. Through kinase phosphorylation, they activate nitric oxide synthase [21] in caveolae where ERs are present, as shown in Figure 3. Caveolae are a particular type of membrane that helps signal transduction through various signaling molecules. ER-α can affiliate with plasma membrane due to receptor palmitoylation [22], but they have no trans membrane domain. When plasma membrane ERs are activated by estrogens, they exist in dimer form [23]. Plasma membrane ER-α attaches with caveolin-1 protein scaffold as well as various signaling molecules such as Src kinase [24], Sch [25], Ras [26], PI3–kinase [27], and G-protein [28]. Non-genomic action of ER is performed by several kinase pathways such as Akt, m-TOR, and MAPK mediated by phosphorylation. In this signaling, ER is activated by epidermal growth factor (EGF), and insulin growth factor (IGF) that initiates further a variety of kinase pathways.

## 4. Small Molecule Inhibitor Targeting Estrogen Receptor-α

In most of the target organs, ER-α exists as a predominant receptor. Consequently, various anti-estrogens (the ligands that block ER-α) were developed to treat breast cancer. Therefore, the primary therapy for ER-α breast cancers is synthetic anti-estrogen drugs. The present review provides a broad summary and discusses the highlights of discoveries, SAR studies, and binding interactions of various degraders of ERs reported to date. 

The X-ray cocrystal structures of ER-α in complex with estradiol (PDB id: 1GWR), is shown in Figure 4 and Table 1. The binding conformation of endogenous ligands or with synthetically developed compounds revealed that amino acid residues Glu-353 and Arg-394 are conserved amino acids residues that participate in hydrogen bond acceptor/donor interactions with estradiol or other ligands such as tamoxifen, raloxifene, diethylstilbestrol, hexahydrocyclopenta[e]chromene, benzothiazine, naphthalene, or indole. As shown in Figure 4, these compounds also utilize amino acid residues such as Asp-351, His-524, and Phe-404 for their binding to the ER-α receptor.

## 5. Strategies for Estrogen Receptor Degradation

The initial rationale of modulating the estrogen receptor that could provide therapeutic benefits in overexpressed ER+ cancers prompted the development of tamoxifen (selective estrogen receptor modulator, SERM), which is widely prescribed as adjuvant therapy following surgery in ER+ breast cancer 14. Later, more attention to selective ER down-regulator (SERD) was provided, which led to the development of fulvestrant, which induces ER degradation. The inclusion of fulvestrant exhibited an improved treatment for patients with ER-positive breast cancers with disease progression following traditional anti-estrogen therapy. The clinical success of fulvestrant showed a promising future for selective estrogen receptor degraders.

A paradigm shift in the way of targeting intracellular proteins by synthetic molecules is expanding the field of medicinal chemistry research. Mechanistically, these synthetic entities make a ternary complex between a protein of interest (POI) and an E3 ubiquitin ligase, leading to the POI ubiquitination and its subsequent degradation through trafficking to the proteasome [47,48]. These chemical entities constitute of (1) a functionality or substructure or small molecular inhibitors that have affinity for POI, (2) a substructure of ligand which has an affinity for recognizing the E3 ligase, and (3) a chemical spacer or linker which tethered (1) and (2). These chemical entities can be called based on their chemical elements PROTACs (proteolysis targeting chimeras), SNIPERS (specific and nongenetic IAP-dependent protein erasers), and degronimers [49]. An illustration flow of the mechanism is displayed in Figure 5. Several successful PROTACs and SNIPERs have been developed in recent years [50,51,52,53].

### 5.1. Peptide PROTACs

Prostate cancer is the second most common cancer in men [55,56], while breast cancer is the most common among women, representing 14% of female cancer deaths [56]. The pathophysiology of prostate and breast cancer demonstrated an androgen receptor (AR) [57] and estrogen receptor-α [58,59] hormonal interventions, respectively. Hormonal and systematic chemotherapies are typically used for metastatic breast and prostate cancer [58,59]. Recent clinical studies revealed: (a) 66% of breast cancer expressed ER-α and 70% respond to hormonal therapy, while prolonged therapy leads to complete refractory cancer even though 30% of refractory tumors show ER-α expression [60], (b) around 85% prostate cancer patients show a good response towards hormonal therapy initially, eventually refractory hormonal cancer within 18–24 months [61,62]. On the other hand, hormonal therapy-resistant cancer shows abrupt signaling where over 50% of cases are linked with molecular and cellular change altering the activation of androgen receptors and cancer cells proliferation even at low serum testosterone levels [63,64,65,66,67,68,69,70,71].

The initial attempt of a Collaboratory work by Sakamoto et al. conceptualized the in vitro activity of estrogen PROTACs and androgen PROTACs [72]. They linked IκBα phosphopeptide to estradiol or dihydrotestosterone (DHT) to recruit ER or AR to SCFβ-TRCP to facilitate their ubiquitination and degradation [72]. These PROTACs consisted of a minimal of 10 amino acid residues where the Serine residue was phosphorylated. Asp-Arg-His-Asp-Phospho(Ser)- Gly-Leu-Asp-Ser-Met was linked to estradiol (in the case of targeting estrogen receptor) or DHT (for targeting androgen receptor), as shown in Figure 6A,B. The in vitro studies of **Protac-2** (that is targeted for estrogen receptor) promotes the concentration-dependent ubiquitination of ER by SCF^β-TRCP^. Initial concentration (0.1–1 µM) showed an onset of ubiquitination, with a maximum degradation (D_max_) achieved at 5–10µM. However, at higher concentrations, a hook effect was observed, which is one of the common limitations of these strategies. In the competent assay, **Protac-2** was tested against IκBα phosphopeptide and estradiol separately. In observation, a 10-times excess concentration of estradiol or IκBα phosphopeptide completely blocked the ubiquitination activity of 1 μm **Protac-2**. In contrast, the simultaneous addition of estradiol and IκBα phosphopeptide failed to replicate the **Protac-2** effect. The addition of a purified yeast 26S proteasome [73] added to ubiquitinated ER (in the presence of SCF^β-TRCP^ and Proatc-2) showed the disappearance of high molecular mass ubiquitin complex within 10 min, illustrating the ER-ubiquitination of **Proatc-2** is proteasomal mediated. The peptidic character of IκBα limits its cellular penetrability and requires phosphorylation on two serine residues to be recognized by SCFβ−TRCP, making it susceptive to phosphatases [74].

In order to overcome such limitations of **Protac-2** and make it in vivo compatible, the same researchers replaced the IKBα peptide with a hydroxyproline-containing pentapeptide derived from hypoxia-inducible factor-1α (HIF-1α). The hydroxyproline-containing pentapeptide is not dependent on phosphorylation and is recognized by the VHL ubiquitin ligase [75,76,77]. Therefore, **Protac-2** was modified into an improved peptide-based PROTAC, **Protac-B**, shown in Figure 6C. Further studies using **MG132** (a proteasomal inhibitor) block the ER degradation, confirming the proteasome-dependent degradation of the estrogen receptor. In cancer cell studies, **Protac-B** inhibits the cell cycle and cell proliferation in hormone-dependent breast cancer cell lines (MCF-7, IC_50_ = 50µM; T47D, IC_50_ = 16 μM), but no effect was observed on hormone-independent tumor cell line (SKBr3) [75].

In 2004 Kim’s research group from the University of Kentucky reported a cell-permeable small molecular proteolysis inducer (SMPI), which utilizes the E3 ligase VHL [78]. The designated SMPI (**E2-SMPI**) constitute two warheads at two terminals, hypoxia-inducible factor-1α (HIF-1α) protein-derived octapeptide motif on one side with a role in recognition of VHL E3 ligase, whereas estradiol on the other side was incorporated to recognize the estrogen receptor, as shown in Figure 6D [78]. HIF-1α plays a critical role under low oxygen stress conditions but is rapidly degraded under normal oxygen levels by the ubiquitin proteasomal pathway. Mechanistically HIF-1α degradation is facilitated by the prolyl hydroxylation of Pro564 (conserved residue), which is also recognized by the pVHL E3 ligase [79,80]. Therefore, adopting a synthetic peptide that contains hydroxyproline residue 564 of HIF-α would prompt a VHL E3 ligase degradation. In this approach, an octapeptide containing residue from 561 to 568 and hydroxyproline at position 564, was found to have enough chemical suitability to be recognized by VHL [79]. When MCF-7 cells were treated with **E2-SMPI** for 15 h, a significant ER level was degraded. In order to confirm the mechanism, an **E2-SMPI** was modified with its octapeptide hydroxyproline, which was replaced with alanine E2-SMPI [ProOH→Ala]. No degradation was observed, justifying the VHL–E2-SMPI mediated degradation of estrogen receptors in MCF-7 cells [78].

Later, the optimization of HIF-α was reported by authors in 2010, where they reduced the length of octapeptide to pentapeptide and further switched the point of linker tethering (C17 to C16 and C7) to estradiol [81]. As shown in Figure 7, the choice of switching the tethering linker points on estradiol structure is based on the following points: (a) C17 in **E2-SMPI** contains an ester linkage which is susceptible to esterase leads to hydrolysis of PROTAC, (b) geldanamycin tethered at C16 of estradiol showed no change in the interaction of estradiol-estrogen receptor [82,83], suggesting C16 position is optimal for binding molecular obese structures such as in this case, (c) C7α position tethering of estradiol also retained its ER binding [84,85,86]. In order to provide further support for C7α position tethering, an E2 derivative with a long hydrocarbon tail at the C7α position (fulvestrant) had reported with potent ER binding affinity. ER degradation studies were performed in MCF-7 cells, where C16α-based **ER-peptide PROTAC-13** and **-14** with the protected pentapeptide were found comparatively potent. Similar to C16α-based **ER-peptide PROTAC-13** and **-14**, C7α-based **ER-peptide PROTAC-24** demonstrated ER degradation levels.

### 5.2. Non-Peptidic PROTACs

Generic lower in vivo stability is one major drawback of peptide-based PROTACs, which led researchers to investigate VHL and CRBN-based E3 ligase ligands. In 2014, GlaxoSmithKline (GSK) patented (WO/2014/108452) PROTACs targeting ER where estradiol was used as an ER warhead, and VHL ligand was used as an E3 ligase warhead. The general structure of PROTACs is shown in Figure 8 [87]. Various PEG linkers were used to conjugate the estradiol and VHL ligand, which resulted in potent VHL-based PROTACs that displayed degradation of ER-α at 1 μM concentration.

In 2019, Wang’s research group from the University of Michigan presented VHL and CRBL-based ER-α PROTACs [88]. Based on the X-ray cocrystal structure [46], authors found that N, N-diethylamino terminal of raloxifene derivative is solvent exposed and can be used as a tethering point for PROTAC strategy (as shown in Figure 9B). PROTACs based on CRBN E3 ligand were found with ineffective degradation activity in the range of 1–1000 nM, whereas VHL E3 ligand-based PROTACs showed ER-α degradation, which led the authors to focus on the VHL-based PROTACs optimization. Based on their previous experience developing bromodomain and extra-terminal (BET) degraders PROTACs, the authors reclaimed the length and chemical composition of the linker is critical to attaining efficient degradation of targeted protein [89,90,91]. Therefore, a series of linker-types with hydrocarbon chain lengths of three-carbons to nine-carbons were incorporated. Quantitative cellular ER-α in MCF-7 cells was measured by Western blotting. The synthesized PROTACs showed similar activities, while **ERD-148** had slightly more potency, which led the authors further to optimize the linker chemical composition of the **ERD-148**. Incorporating oxygen atoms in these linkers resulted in PROTACs showing effective ER-α degradation-inducing ability at 10 and 100 nM in MCF-7 cells. In addition, **ERD-148** had limited aqueous solubility because of the incorporation of long hydrocarbon chain hydrobophicity, which was improved by incorporating heteroatoms (oxygen) in the hydrocarbon chain. **ERD-308** was yielded as the most potent VHL ligand-based ER-α PROTAC. The induced ER degradation of **ERD-308** in T47D ER+ breast cancer cell line was found DC_50_ = 0.43 nM, while maximum ER-degradation (>95%) was noted at 5 nM, but at a higher dosage (at 1 μM) hook effect was observed. Kinetic studies of ER degradation of **ERD-308** was investigated in MCF-7 cells. At 30 nM, more than 80% ER protein cellular level was reduced in first 1 h after treatment, and a complete ER degradation was achieved at the 3 h time-point. In contrast, fulvestrant, a typical SERD, had shown a mild effect on ER cellular level at 1 h and attained maximum ER degradation (90%) after 24 h of treatment. Similar kinetic data was also obtained for **ERD-308** and fulvestrant in T47D cells. Based on the observation in both the cell lines, the author concluded that **ERD-308** as a fast ER-α degradation kinetics. Furthermore, the ER-αprotein degradation mechanism was studied by using competitive assays. In those assays, a significant reduction in degradation activities of **ERD-308** (at 30 nM) were noticed with the addition of raloxifene (1 μM) and 1 μM of the proteasome inhibitor (carfilzomib). Adding VHL ligand (5 or 10 μM) also reduced the degradation activity of **ERD-308**. These studies showed that ERD-308 ER degradation activity is proteasomal mediated. The effect on cell proliferation of **ERD-308** was studied in MCF-7 cells using WST-8 cell proliferation assay, with raloxifene and fulvestrant used as controls during the experiments. **ERD-308** showed IC_50_ (0.77 nM) and a maximum inhibition Imax of 57.5%. In contrast, fulvestrant was also found quite potent in the inhibition of cell proliferation with Imax = 43.8%, while raloxifene was only able to achieve an Imax value of 34.0%. However, as anticipated, the authors didn’t find not cell proliferation inhibition of **ERD-308** in triple-negative breast cancer cell (MDA-MB-231) and primary human mammary epithelial cells. Using quantitative reverse transcription-polymerase chain reaction analysis, **ERD-308** downregulates the ER-regulated genes (*pGR* and *GREB1*) [88].

As clinical studies revealed that somatic mutations (Tyr537Ser (Y537S) and Asp538Gly (D538G)) in estrogen receptor α (ER-α) ligand-binding domain (LBD) are detected in ~30% of endocrine-resistant metastatic ER-positive breast cancer patients [92]. These ESR1 mutations lead to the agonist conformation of ER-α, confer an estrogen-independent phenotype, and aggravate the resistance to antiestrogen drugs [93,94,95,96]. In order to investigate the effectivity of current developed PROTACs, Wang and Rae Coworkers studied the ER-α degradation in estrogen-independent clones engineered MC7-cells that were expressing the ESR1 LBD mutations Y537S and D538G [97]. The most potent PROTAC resulted from their previous study [88], **ERD-148** superiorly downregulated the ER-α expression in compared to fulvestrant in wild-type MCF-7 and CRISPR/cas9 knock-in LBD mutated MCF-7 cells. The antiproliferation cellular effect of **ERD-148** was reversed by 17β-estradiol treatment, suggesting a reversible competition inhibition of **ERD-148,** showcasing its ER antagonism. Importantly, **ERD-148** showed minimal non-specific toxicity in estrogen-independent cell lines (MDA-MB-468 and MDA-MB-231 cells) at concentrations above its ~IC90, indicating its promising therapeutic role in ER-positive cancers [97].

## 6. SNIPERs

SNIPERS work similarly to PROTACs, but because they utilize different E3 ligase proteins, and therefore called IAP-based PROTACs (or, more specifically, SNIPERs). SNIPERs utilize IAPs (Inhibitors of apoptosis protein) proteins, which are a class of negative regulators for apoptosis in mammalian cells and thereby inhibit cellular apoptosis by inhibiting the caspase cascade. There are eight members in the mammalian IAP family: cellular IAP1 (c-IAP1), X-chromosome-linked IAP (XIAP), cellular IAP2 (c-IAP2), neuronal IAP (NAIP), ubiquitin-conjugating BIR domain enzyme (BRUCE), survivin, IAP-like protein 2 (ILP2) and melanoma IAP (ML-IAP), as shown in Figure 10 [50,98,99]. These proteins are a family of antiapoptotic proteins, which have one-to-three baculoviral IAP repeat (BIR) domains that are of a 70–80 amino acids long motif, as shown in Figure 10 [100,101,102,103]. Along with BIR domains, IAPs also contain a ubiquitin-associated (UBA) and a RING domain in their structure, as shown in Figure 10. The BIR2 domain of XIAP prevents the activation of caspase-3 and caspase-7, while the BIR3 domain selectively prevents the activation of caspase 9 [104]. Interestingly, c-IAP1 and c-IAP2 also interact with caspase-3/9 but don’t inhibit their activation [105]. However, c-IAP1 and c-IAP2, along with XIAP, possess one UBA domain that contributes to ubiquitin chain binding [106]. ILP2 possesses RING finger domain, which assists its interaction with E2 ubiquitin-conjugating enzymes (UBCs), while BRUCE possesses a UBC domain. In addition, IAPs do their ubiquitylation and associated proteins. cIAP1, CIAP2, and XIAP (X chromosome-linked IAP) are directly involved in regulating the caspases cascade and are extensively studied IAPs. IAPs became attractive targets in devising a novel cancer-targeting approach as reports suggested their overexpression in multiple malignant cancers commonly related to cell proliferation, drug resistance, and poor prognosis [107,108,109]. The retrieved chemical biology of IAPs was exploited in developing their potent small molecule inhibitors/antagonists as potent anticancer therapeutics, while some of them succeeded in reaching the various stages of clinical trials, as shown in Figure 11 [110,111,112].

In addition to their identifications as cancer targets, they also show E3 ligase activity, which thrives PROTAC development. Bestatin (BS) and its derivative show binding activity towards the BIR3 domain of IAPs and therefore induce self-ubiquitination-based proteasomal degradation [113]. With such high affinities for IAPs, BS-based compounds (as shown in Figure 12A) were an easy choice for chemical biologists to adapt their structure to derive a class of SNIPERs. Natio et al. reported that MeBS could enhance the chemosensitivity of cancer cells toward apoptosis induced by cancer drugs. Mechanistic studies revealed that MeBS induces RING-dependent auto-ubiquitylation and proteasomal degradation of cIAP1 by interacting with the third BIR domain of cIAP1 (as shown in Figure 12B). The SAR studies revealed that reduced cIAP1 levels were maintained even with the replacement of methyl carboxylate with bulkier groups, indicating no participation of the methyl carboxylate group of MeBS during its interaction with cIAP1. Although, structural changes in the bestatin backbone severely reduced the activity by lowering the cIAP1 expression, suggesting an active role of bestatin backbone in interaction with cIAP1 [113]. These observations prompted Natio et al. to extend the molecular frame to devise SNIPERs, where the position of the methyl group is used as tethered to attach a linker which is further tethered to molecules that have intrinsic activity for targeted protein, as shown in Figure 12C.

### 6.1. Non-Peptidic SNIPERs

In 2011, Hashimoto’s research group from the University of Tokyo (Japan) utilized the Estrodiol -based SNIPERs to degrade the estrogen receptors [114]. The authors used an estrone scaffold to functionalize it at the C17 position with bestatin (a ligand of cellular inhibitor of apoptosis protein 1 (cIAP1), one of the IAPs) through a linear linker [114]. Instead of calling them SNIPERs, authors choose to call them **PROTAC-11** and **PROTAC-12**. Western blotting estimated the ER levels in MCF-7 cells for **PROATC-11**, estrone, bestatin, and **PROTAC-12** (a negative control that lacks chemical specificity to bind with cIAP1). Among these, **PROTAC-11** showed a comparatively low level of ER cellular level. However, in our opinion, the imine functionality with oxygen (marked in a red dotted circle in Figure 13) makes **PROTAC-11** susceptible to hydrolysis at physiological pH; therefore, PROTACs with these kinds of functionalities would have a shorter half-life and limited physicochemical properties.

In 2012, Naito and Kurihara, co-workers from the National Institute of Health Sciences Tokyo, Japan, developed SNIPERs (original publication: **PROTAC-5**, **PROTAC-6**, and **PROTAC-7**) constituting two warheads: 4-hydroxytamoxifen, that recognize the ER while bestatin moiety recognizes cIAPs [115]. Using an X-ray cocrystal (PDB id: 3ERT) N, N dimethylamino moiety of 4-hydroxytamoxifen with estrogen receptor, an alkyl tethering was used to join the bestatin moiety with the N, N dimethylamino moiety of 4-hydroxytamoxifen, as shown as red circle in Figure 14A,B [115]. The authors studied the dose-response effect on ER-α protein degradation of **PROTAC-5**, **6**, and **7** on MCF-7 breast cancer cells. By comparing the data produced by Western blotting, ER-α protein was measured in the MCF-7 cells. The protein ER-α levels were increased with (E/Z)-endoxifen treatment [116], and reduced levels of ER-α were noticed with 10 or 30 μM of compound **5, 6**, or **7** in MCF-7 cells. However, when cells were treated with (E/Z)-endoxifen and methyl bestatin) together, no change in ER-α protein degradation was observed [113], suggesting the PROTAC strategy when (E/Z)-endoxifen tethered with methyl bestatin (at a concentration of greater than 10 μM) efficiently degrades the cellular ER-α protein. In order to provide further evidence, a competitive assay against a proteasome inhibitor, **MG132**, abrogated the cellular ER-α protein degradation induced by **5**, **6**, and **7**, demonstrating the proteasomal-mediated degradation of ER-α protein [115]. In our opinion, this study would be more appealing to chemical biologists if the authors had compared the effect of configurational isomerization of these SNIPERs on ER-α cellular degradation, as specific isomers could be more potent than reported values.

In order to understand the ER-α degradation mechanism of **SNIPER(ER)-3** [117] (reported initially in a previous publication as **PROTAC-7** [114]), the authors performed extensive biochemical studies [117]. In their study, bestatin ((−)-N-[(2S, 3R)-3-amino-2-hydroxy-4-phenyl-butyryl]-L-leucine) was used as cIAP ligand, which in methyl ester form can directly interact with BIR3 domain of cIAP1 and triggered auto-ubiquitylation of cIAP1 and subsequently proteasomal degradation of cIAP1 [113]. SAR studies suggested that the methyl ester version of bestatin (MeBS) degrades cIAP1 irrelevant to the methylation even onto the other residues [113]. In order to determine the specificity of these synthetic’s SNIPER-induced ER-α degradation from the cellular degradation (which takes place after the estrogen binding to its receptor [118]), the authors used MCF-7 cells precultured in estrogen-depleted serum-containing media for 96 h. Later, treatment with **SNIPER(ER)-3**, β-estradiol, 4-OHT, MeBS, and a combination of MeBS and 4-OHT were performed. By using Western blotting, in the absence of β-estradiol, **SNIPER(ER)-3** showed reduced cellular levs of ER-α protein, while no effect ER-α levels were observed in the presence of 4-OHT, MeBS, or with their combination (4-OHT, MeBS). Furthermore, cellular levels of cIAP1 were also decreased by **SNIPER(ER)-3** (at 30 μM) as well as MeBS, indicating the **SNIPER(ER)-3** triggered the auto-ubiquitylation and proteasomal degradation of cIAP1. In addition, a decrease in ER-α cellular level was observed in the presence of β-estradiol. β-estradiol-induced expression of pS2 mRNA but not the **SNIPER(ER)-3**, while **SNIPER(ER)-3** inhibit the upregulation of pS2 induced by β-estradiol (same as 4-OHT did). These mRNA expression studies suggested **SNIPER(ER)-3** induced ER-α degradation is independent rather than activation of ER-α receptor, exemplifying an example of event-driven pharmacology [117].

In order to achieve higher SNIPER effectivity, a collaborative work by Naito co-workers in 2017 showed utilization of other IAPs protein ligands (as shown in Figure 14D,E). As SMAC/DIABLO is an endogenous protein partner of IAPs that behaves as an antagonist. It utilizes its N-terminal IAP binding motif to interact with IAPs. By ligand-based design, the binding conformation of IAP-binding tetrapeptides of SMAC, and new peptidomimetics as IAP antagonists were developed. These peptidomimetics showed potent effects and cell permeability and are in the clinical development phase [119,120]. These IAP antagonists show a strong affinity toward the BIR domains of IAP proteins than bestatin, which triggers a rapid autoubiquitylation, followed by proteasomal degradation of IAPs [121,122,123], and, therefore, well-suited to improve the predesign of **SNIPER(ER)-3** (or **PROTAC-7** [114]). As warheads of estrogen receptors, the following ligands (4-OHT, Raloxifene, estradiol, lasofoxifene) were used, while **MV1** and **bestatin** were used as IAPs warheads, as shown in Figure 14D. **SNIPER(ER)-19** (4-OHT-MV1, which had 4-OHT as estrogen receptor warhead and MV1 as IAP warhead) at 30 nM decreased the ER-α levels in MCF-7 breast cancer cells. To expand the scope of SNIPERs, various ER-α ligands were combined with IAP antagonists using different length linkers, where **SNIPER(ER)-20** had the most potent activity. To achieve a more potent activity, other IAP antagonists were incorporated, which led to the identification of **SNIPER(ER)-87**. **SNIPER(ER)-87** (DC_50_ =< 3 nM; D_max_ ≈ 100 nM) is derived from **LCL161**, showed a more ER-α reduction than **SNIPER(ER)-20**. The effect of the length of linker also played a role, as a longer linker containing **SNIPER(ER)-87**, **SNIPER(ER)-88**, later one had ER-α degradation within 1 h after treatment and sustained for 48 h, while shorter length linker containing **SNIPER(ER)-89** had an attenuated degradation activity. In the competent assay, estrogen receptor warhead (4-OHT) and the IAP antagonist (**LCL161** derivative), when treated together, no decrease in the ER-α protein level was observed in MCF-7 breast cancer cells (IC_50_ = 15.6 nM), suggesting **SNIPER(ER)-87** as an ideal example of event-driven pharmacology. **SNIPER(ER)-87** prevented the proliferation of ER-positive breast tumor cells (MCF-7, IC_50_ = 15.6 nM; T47D, IC_50_ = 9.6 nM) by suppressing the ER-dependent transcriptional activation. Similar to other SNIPERs and IAP antagonists, **SNIPER(ER)-87** reduced the cIAP1 level and, to some extent, levels of XIAP, suggesting an autoubiquitylation, followed by proteasomal degradation of cIAP. ER-α degradation proficiency of **SNIPER(ER)-87** was measured in vivo using ovarian ER-α level. Later, **SNIPER(ER)-87** (10 or 30 mg/kg body weight) was injected intraperitoneally into the female BALB/c mice, where a significant reduction of ER-α protein levels in the ovary was recorded. Later, to evaluate in vivo ER-α knockdown in a tumor model, authors developed MCF-7 breast tumor xenografts in nude mice, where **SNIPER(ER)-87** deceased the ovarian cellular levels of ER-α.

Encouraged by the proficiency of **SNIPER(ER)-87**, the authors screened other IAP antagonists in search of gaining more ER-α degradation potency [124]. Therefore, the authors used IAP antagonists from reported studies [125,126] and screened them for their binding affinity to XIAP to decrease the ER-α expression in MCF-7 cells, as **SNIPER(ER)-87** recruits XIAP for ER-α degradation [127]. As shown in Figure 15, replacing LC131 with other IAP antagonists yielded five (SNIPER(ER) (**-105**, **-110**, **-113**, **-119**, **-126**) (as shown in Figure 15) that were either comparable or more potent than **SNIPER(ER)-87** for durations of 4 h and 48 h, respectively, as shown in Table 2 [124]. **SNIPER(ER)-130** and **-131** demonstrated an abrogated activity despite their potent XIAP affinities, which indicates that increased XIAP binding is vital to attain an effective ER-α degradation but not the only determinant to achieving complete degradation activity. However, (**SNIPER(ER)-104**, **-118**, **-121**, **-134**, and **-136**) had not triggered ER-α degradation from 1 nM until 100 nM and exhibited lower binding affinities to XIAP, as shown in Table 2. While **SNIPER-105**, **-110**, and **-126** demonstrated more potent ER-α degradation than **SNIPER(ER)-87** in ER-α-dependent breast cancer cell lines (T47D and ZR-75-1). Studied IAP antagonists of the current study (as shown in Figure 15) were reported to induce proteasomal degradation of cIAP1 and their respective targeted proteins [117,127,128,129,130,131,132]. An increasing trend in the binding cIAP1 affinity of **SNIPER(ER)-105**, **-110**, **-113**, **-119**, and **-126** proportionally decreases the cIAP1 levels compared to **SNIPER(ER)-87** (as shown in Table 2). Further analysis showed 48 h exposure prolonged exposure potently reduces cellular levels of XIAP in MCF-7 cells compared to **SNIPER(ER)-87**. Similarly, prolonged exposure of 48 h, a pronounced reduction in the cellular level of XIAP with these SNIPER(ER)s in T47D but weaker after 4 h of exposure. The differential degradation of cIAP1 and XIAP exhibited by SNIPER(ER)s, indicates a degradation-specific mechanism for these IAPs. As authors previously observed ER-α degradation of **SNIPER(ER)-87** preferentially utilized XIAP [127]. To scrutinize IAP-specific ER-α degradation of these SNIPER(ER)s, (**SNIPER(ER)-105**, **-110**, and **-126**) were tested in MCF-7 and T47D cancer cell lines with or without **MG132**. The authors observed the anti-ER-α precipitates contained both IAPs (XIAP and cIAP1) in the SNIPER(ER)s treated cells, while cIAP2 cellular levels were not examined as MCF-7 and T47D cells do not express cIAP2 under typical cell culture and cIAP1down-regulated conditions. By comparing the cellular XIAP levels with cIAP1 in cell lysates, the authors observed preferential recruitment of XIAP to ER-α of these SNIPER(ER)s. To consolidate XIAP recruitment-based ER-α degradation, authors depleted the XIAP using siRNA in MCF-7 and T47D cells, which results in suppresses the ER-α degradation induced by SNIPER(ER)s. These observations led authors to consider the **SNIPER(ER)-105**, **-110**, and **-126** as preferential XIAP recruiter to ER-α, which in contrast is similar to **SNIPER(ER)-87** illustrated mechanisms. Interesting, similar molecular structure containing SNIPERs (regioisomers), which only differ in their arene substitutions (highlighted in red color in Figure 15) of IAP antagonist, **SNIPER(ER)-110** (*meta*-substitution), **-113** (*para*-substitution), and **-119** (*ortho*-substitution). Meta-substituent version (**SNIPER(ER)-110**) exhibited higher ER-α degrade activity which led the authors to choose **SNIPER(ER)-110** to study the biochemical mechanism. When these synthesized SNIPERS tested against ER-α-positive breast tumor cells (MCF-7 and T47D) and ER-α-negative breast tumor cells (MDA-MB-231), a significant number of MCF-7 cells, but not T47D cells were underwent apoptosis under microscopic analysis, which later confirmed by flow cytometric analysis and annexin V/propidium iodide (PI) staining. These observations suggested that **SNIPER(ER)-105**, **SNIPER(ER)-110**, and **SNIPER(ER)-126** (at above 50 nM) induce higher apoptosis in MCF-7 cells than **SNIPER(ER)-87**, but reasonable apoptosis was not observed in T47D cells [124].

To comprehend the reasoning behind differential apoptosis induced by these **SNIPER(ER)-105**, -110, and -126 in MCF-7 cells, but not in T47D cells, and these SNIPERs induce degradation of cIAP1 and XIAP more potently than **SNIPER(ER)-87**, authors carefully examined the effects of IAP depletion in MCF-7 and T47D cells. Silencing of cIAP1 and XIAP by siRNA somewhat reduced the number of MCF-7 cells, while combined silencing (cIAP1 and XIAP) led to nearly 50% decrease in MCF-7 cell number when compared with untreated cells and was accompanied by caspase activation. While in T47D cells, similar IAPs silencing, no observation of reduced cell number and caspases activation was made. Interestingly, silencing of ER-α in MCF-7 and T47D cells led to decrease the cell numbers in both cell lines, without caspase activation. Additionally, triple silencing of ER-α, XIAP, and cIAP1 led an MCF-7 cell number reduction higher than what was observed with T47D cells; and caspase activation on observed in MCF-7 cells. In contrast, an observation of LCL161 sensitized tumor necrosis factor α (TNFα)-dependent apoptosis in MCF-7 cells, but not T47D cells was made. Based on these observations, the authors advocate IAPs dependency behind MCF-7 cells survival, but not for T47D cells; characterizing IAPs involvement in the selective apoptosis induction in MCF-7 cells treated to **SNIPER(ER)-105**, **-110**, and **-126**. However, if an IAP ligand of higher binding affinity is incorporated into SNIPER, that could induce a significant level of protein knockout and cytotoxicity to cancer cells requiring IAPs for their survival; these differences can be subsided or largely minimized. Therefore, these observed differences in those two cell lines could be resulted from their different needs for IAPs.

Based on their previous ER-PROTAC experience, GSK patented (WO/2016/169989A1) SNIPERs using raloxifene derivative [135] and IAP recruiting moiety in 2016, as shown as Figure 16 [136]. These SNIPERs exhibited more than 50% ER-α degradation below 1 μM concentration.

### 6.2. Peptide-Based SNIPERs

In 2016, Demizu et al. from the National Institute of Health Sciences (Tokyo, Japan) developed peptide-based SNIPER for estrogen receptor α (ER-α), where MV1 structure was used to recruit IAP (cIAP1/cIAP2/XIAP) to trigger the ubiquitylation of ER-α [137]. The rationality behind using a peptide instead of an ER-α-based small heterocyclic molecule, is small ER-α inhibitor inherited with a weaker to moderate agonistic effect on ER-α in uterine cancer cells, which can precipitate the risk of endometrial cancer [138,139]. Therefore, in search of newer scaffolds for ER-α inhibitors, peptidomimetic Estrogen Receptor Modulators (PERM) were identified. The authors utilized the PERM3-R7 as ER-α warhead and MV1 ligand structure for recruiting the cIAP1. PERM3-R7 is designed on a PERM3 scaffold, which itself is a LXXLL-like mimetic (where L represents leucine, while X represents other amino acid residues) of the Steroid Receptor Coactivator 1 (SRC-1) that binds with the ER-α surface [40,140], while R7 represents a hepta-arginine fragment that was fitted to PERM3 to enhance the cellular permeability of PERM. In addition, the peptide nature could allow PERM3-R7 to adopt a stable α-helical conformation to prevent ER-based transcription and suppresses the expression mRNA of pS2 (an ER-mediated gene whose expression is upregulated by E2) at the cellular level [141,142].

Based on Western blotting, the estrogen receptor cellular levels were quantified in MCF-7 cells. Repetition of PEG linker was used to conjugate the PERM-R7 and MV1 ligand structure, no ER degradation activity for the shorter (**PERM-R7 SNIPER-2**) and medium PEG linker-based peptide SNIPERs (**PERM-R7 SNIPER-3**), as shown in Figure 17. Meanwhile, a longer PEG linker (**PERM-R7 SNIPER-4**) showed dose-dependent ER-degradation activity and reduced levels of cIAP1, as shown in Figure 17. Interestingly, when PEG linkers were replaced with five β-alanine residues, yielded **PERM-R7 SNIPER-5** showed comparatively lower ER-degradation **PERM-R7 SNIPER-4** (also showed cytotoxicity effect at >6.0-μM) but showed no cytotoxicity effects even at higher dosage (20 μM). Furthermore, the ER degradation for these peptide SNIPERs were found to be proteasomal mediated as **MG132** abrogated their degradation activity. Interestingly, an observation of the reduced cellular levels of cIAP1 protein with 1.0 μM **PERM-R7 SNIPER-3**, suggested that these peptide SNIPERs activate the autoubiquitylation, followed by proteasomal mediated cIAP1degradation itself with a pan-IAP ligand (MV-1) as a recruiting warhead, therefore resulted activities could be cumulative of mixed biochemical events that took place at cellular levels.

## 7. Antibody Drug Conjugates Based PROTACs and SNIPERs

Research collaboration of Genentech teams from the United states of America (South San Francisco, CA 94080, USA) and China (Shanghai 200131) developed antibody conjugates with their own *in-house* estrogen receptor degraders [143]. Researchers used a tamoxifen scaffold (in a 1:1 mixture of E and Z-isomers) for preparing these degraders [144,145], as enlisted in Table 3. Based on the ligase choice, the **5** and **8** were SNIPERs, as shown in Figure 18A. Using immunofluorescence (IF) readout in wild-type MCF7 cells and engineered MCF7 cells (that had an over-expressed HER2 onto the cell surface), **SNIPER-5** and **VHL-PROTAC-6** showed a nearly complete ER-α protein degradation [146]. Modification with an epimer of hydroyproline in VHL structure of **VHL-PROTAC-6** led to the formation of **VHL-PROTAC-7** (as a negative control), as hydroxylation of proline of VHL abolishes the adopting conformation binding of such modified VHL structures, as shown in Figure 18B.

Interestingly, **VHL-PROTAC-7** showed a weak but measurable ER-α degradation activity compared to its parent molecule **VHL-PROTAC-6**. To rationalize such observation, the researcher when tested the exact ER-α warhead (endoxifen, a metabolite of tamoxifen), showed ER-α degradation effects. However, the authors point out that the degradation consistency with **VHL-PROTAC-7** and endoxifen (ER-α warhead of **VHL-PROTAC-7**), could be resulted from their binary complex formation between ER-α protein itself with PROTAC through endoxifen (as ER-α warhead). In our understanding, to demonstrate these presuppositions, authors need to carry out extensive concomitant experiments with other VHL ligands (which can work as negative control) or molecular probes that can work as ER-warheads.

Later, the quantitative Western blotting analysis showed concentration-dependent ER-α degradation in MCF7-neo/HER2 cells of **VHL-PROTAC-6**. Interestingly a discrepancy in the degradation between immunofluorescence and Western blotting methods for **VHL-PROTAC-7** and endoxifen was observed. These differences could be originated from weak ligands triggered changes in ER-α conformational and/or subcellular localization, and that led to minimize/restrict the recognition of the ER-α detection antibody in the intracellular microenvironment during immunofluorescence experiments (this issues would be subsided as cells were lysed after treatment and ER-α protein is fully denatured in Western blotting method) [147]. Authors developed a control (**SNIPER-8**) based on **SNIPER-5** structure, where O-benzylation was carried out onto the phenol of tamoxifen of **SNIPER-5**, as shown in Figure 18A. **SNIPER-8** displayed a poor DC_50_ value compared to its parent (**SNIPER-5**), showcasing relevance of phenolic (-OH) of tamoxifen for its potency, which in our opinion can be attributed either through physiochemically (by improving polarity) or biophysically (involved in H-bond acceptor or donor interaction with the target protein).

As the metabolic stability of antibody conjugates is feared to be their main limitation for their in vivo application among chemical biologists, authors evaluated the **SNIPER-5** and **VHL-PROTAC-6** towards human liver lysosomes. In this study, the authors assess their metabolic survivability when exposed to the intracellular lysosomal environment [148]. **SNIPER-5** and **VHL-PROTAC-6** exhibited reasonable in vitro stability (T_1/2_ > 24 h), where a fraction of amide linkage hydrolysis of **VHL-PROTAC-6** was observed. In order to achieve desired antibody-conjugated degrader, authors initially connect a linker (Valine-Citrulline-para-amino-benzyloxy) to **SNIPER-5** to yield **10** (a linker drug molecule). Such linkers are key modalities commonly employed in bioorthogonal chemistry as they tend to undergo protease-mediated cleavage in lysosomes, proficiently unloading the drugs following intracellular antibody delivery [149,150]. In addition, maleimide functionality at the terminus of these linkers quickly reacted with engineered cysteine residues of monoclonal antibodies, yielding the required antibody conjugates [151,152,153,154]. However, when authored attempted to conjugate **10** with an HER2-targetted antibody that contained an engineered cysteine at LC-K149 site [target drug-antibody ratio (DAR) = 2.0], a high degree of self-aggregation behavior was observed for formed antibody conjugate [155]. Such behavior of high degree of self-aggregation might be attributed by the high hydrophobic character of linker-drug (**10**) intolerable on the antibody surface [156]. Therefore, purification procedures were followed to attain enough pure quantity of antibody conjugate for biological testing. Interestingly, the choice of a site on mAB was reported to afford in vivo stable maleimide-derived ADCs that don’t show retro-Michael-related deconjugation [151,157,158].

In order to rectify such physiochemical limitations with antibody conjugates, the authors changed the linker tethering point to the phenolic head of tamoxifen of **SNIPER-5**, as shown in Figure 19A. In rationality, the para-location of the electron-withdrawing group assists in releasing the **SNIPER-5** during protease-mediated cleavage of **11** (another linker drug molecule similar to **10**). To the authors’ surprise, no substitution of the basic amine of **SNIPER-5** was observed. Therefore, this basic amine could be used as a physiochemical handle by the authors to protonate it, increase the aqueous solubility, and decrease the self-aggregation of final antibody conjugates [159]. Keeping these points in mind, the authors prepared a HER2-targeting DAR2 conjugate (**HER2-11-lc**) which had shown lesser self-aggregation properties. **HER2-11-lc** displayed potent ER-α degradation in MCF7-neo/HER2 cells and far more pronounced than the same linker used B7H4-targeting control conjugate (**B7H4-11-lc**) and an unconjugated HER2-targeting mAb, as shown in Table 4. The ER-α degradation activity reduces when **HER2-11-lc** ADC used in wild-type MCF7 cells, which possesses lesser expressed HER2 receptors on their surface. Later, when authors studied in vivo pharmacokinetic stability of **HER2-11-lc** ADC, where an amide linkage present in the linker of **11** underwent biotransformation after 96 h (intravenous route administration). These outcomes led authors to explore ADCs devoid of the structure of **SNIPER-5**, as **11** (linker drug molecule) was a derivative of **5**, and prepared VHL-PROTACs based linker drug molecules (**12** and **13**) from **VHL-PROTAC-6** and **VHL-PROTAC-7**, respectively, as shown in Figure 19. The derived linker conjugates of **VHL-PROTAC-6** and **VHL-PROTAC-7**, **12**, and **13** had a methanethiosulfonyl (MTS) disulfide functionality that facilitates the reactive elimination with surface-exposed cysteines of engineered monoclonal antibodies, producing desired ADCs [160]. Based on the literature, upon cellular internalization of these ADCs, lysosomal antibody catabolism, followed by disulfide reduction and self-immolation to release **VHL-PROTAC-6** and **VHL-PROTAC-7** [161,162]. Several ADCs were derived from these VHL-PROTACs based linker drug molecules (**12** and **13**). Meanwhile, significantly lower hydrophobic character of these VHL-PROTACs based linker drug molecules (**12** and **13**) than SNIPERs based linker drug molecules (**10** and **11**) prompted authors to increase conjugation of **12** and **13** with six surface cysteines containing mAbs, and achieved without tackling a high degree of self-aggregation [target drug-antibody ratio (DAR) = 6.0]. Compared to the ADC containing DAR2 (derived from **SNIPER-5**), newer ADCs would produce more intracellular ER-α degrader release following ADC-mediated delivery [163,164,165]; as well as the newer location of engineered cysteines were introduced (LC-K149, HC-L174, and HC-Y373) [155], that would be capable of producing disulfide-linked conjugates with relatively good in vivo stability [157,158].

The ADC (**HER2-12**, ER-α DC_50_ = 0.04 ± 0.007 μg/mL, Dmax = 99%) resulted from **12** exhibited a potent ER-α degradation, comparatively more pronounced effect than its respective unconjugated **HER2 mAb** (ER-α DC_50_ = 0.04 μg/mL, Dmax = 43%) in MCF7-neo/HER2 cell, as shown in Table 4. In addition, a milder ER-α degradation for the control conjugate (**CD22-12**) was observed. However, comparatively, the degradation activity difference of **CD22-12** (ER-α DC_50_ = 0.51 ± 0.094 μg/mL, Dmax = 90%) and **HER2-12** (ER-α DC_50_ = 0.04 ± 0.007 μg/mL, Dmax = 99%) were less pronounced when compared to significant difference as observed with previously developed conjugates in the same study (**HER2-11-lc** (ER-α DC_50_ = 0.11 ± 0.001 μg/mL, Dmax = 81%) and its respective control conjugate, **B7H4-11-lc** (ER-α DC_50_ = 50 ± 23 μg/mL, Dmax = 40%)) in MCF7-neo/HER2 cell. As these outcomes for **HER2-12**, **HER2 mAb**, **CD22-12** were received from immunofluorescence, Western blotting was used for cross-validation. Similar to previously reported disulfide-containing linker conjugates [146], the disulfide linker of **HER2-12** and **CD22-12** exhibited in vitro partial hydrolyzed in experiments. Interestingly, these ADCs are known to have in vivo stability and preferential pharmacodynamics for targeted versus non-targeted effects in xenograft tumor models [146], which prompted authors for further investigation. A similar ER-α degradation profile in MCF7-neo/HER2 cells was observed for control conjugate, **HER2-13** (DC_50_ = 0.05 ± 0.016 μg/mL; D_max_ = 87%) compared to **HER2-12** ADC (DC_50_ = 0.04 ± 0.007 μg/mL; D_max_ = 99%). However, different activity was noted in wild-type MCF-7 for **HER2-13** (DC_50_ = 0.70± 0.068 μg/mL; D_max_ = 51%) compared to **HER2-12** ADC (DC_50_ = 0.23 ± 0.007 μg/mL; D_max_ = 95%). The significant ER-α degradation differences between **HER2-12** and **HER2-13** in wild-type MCF-7 cells which don’t express HER2 receptors, reasonably (a) nominal ER-α degradation of **HER2-13** in wild-type of MCF-7 cells, is due to possession of endoxifen-like structure (**VHL-PROTAC-7** versus endoxifen), (b) increases effect of **HER2-12** compared to **HER2-13** because of additional VHL-mediated degradation (**VHL-PROTAC-6** versus **VHL-PROTAC-7**). Relative potency of **HER2-13** in MCF-7-neo/HER2 compared to wild-type MCF-7 apparently because of the synergistic effect of endoxifen-like structure activity as well as **HER2-mAb** ER-α alterations (as observed by the authors). Due to these combined effects, accessing the intracellular release kinetics from **HER2-12** in MCF7-neo/HER2 experiments would be difficult.

The authors also explored the conjugation of **VHL-PROTAC-6** to antibodies using alternative linkers other than disulfide-based linkers. They used pyrophosphate di-ester containing linker, which maleimide on one side that could be exploited to conjugate antibodies. The pyrophosphate di-ester functionality was used to functionalize the hydroxyproline moiety of VHL ligand of **VHL-PROTAC-6**, to yield **14** (another linker drug molecule). The yielded ADC must follow a sequential phosphodiesterase activity, phosphatase-mediated hydrolysis, and lysosomal antibody catabolism to release **VHL-PROTAC-6**, as shown in Figure 20. The choice of pyrophosphate di-esters was based on their successful integration in ADCs containing glucocorticoid payloads [166,167] but also reported to be underutilized by the submaximal enzymatic activity of phosphatase enzymes [166]. Therefore, before proceeding to yield the ADC of **14** (**HER2-14**), the authors study the corresponding cleavage of secondary phosphate functionality in a lysosomal environment, as shown in Figure 20.

Interestingly, **HER2-14** displayed a potent ER-α degradation activity in MCF7-neo/HER2 cells compared to its respective anti-CD22 ADC and the unconjugated HER2 mAb (as shown in Table 4). Efficient intracellular delivery of **VHL-PROTAC-6** undergoes the expected phosphodiesterase cleavage mechanism. As the authors didn’t synthesize the pyrophosphate di-ester control conjugate derived from **VHL-PROTAC-6**, therefore were not able to conduct a study to differentiate the mechanistic study; therefore, this activity could also be the outcome of a combined effect due to the possession of endoxifen-like ligand in the structure and **HER2-mAb** effects. To the authors surprise, **HER2-14** displayed comparatively more potent ER-α degradation in wild-type MCF7 cells (DC_50_ = 0.09 ± 0.013 μg/mL, DC_max_ = 95%) than its similar ADC (**HER2-12**, DC_50_ = 0.23 ± 0.007 μg/mL, DC_max_ = 95%) that did not over-express the HER-2 surface receptor. In our opinion, it would be more appealing for chemical biologists to study the cleavage studies of **HER2-12** and **HER2-14** as both structures are chemically the same except for their conjugate linker-type differences, where the former used disulfide linker while the later one used pyrophosphate di-ester linker.

## 8. Photocaging PROTACs of Estrogen Receptor

Gaining precise control over the biological activity of smaller-sized probes has always interested chemical biologists and medicinal chemists. However, a light-controlled higher spatiotemporal resolution has been exploited as chemical biology tools [168] and in phototherapies [169], where a specific wavelength of light activates the bioactive molecule [170,171].

Deiter’s research group from the Department of Chemistry, University of Pittsburgh, Pittsburgh, Pennsylvania, USA, developed a coumarin-based photocaged VHL ligand. Initially, the author investigated an X-ray co-crystal structure (PDB id: 4W9C) to find a tethering point for a photocleavable group onto the VHL ligand [172]. As shown in Figure 21A, the author noticed hydroxyproline moiety buried into the binding cleft and had H-bond interactions with Ser111 and His115, which are critical amino acid residue interactions to recognize VHL by HIF1-α protein [173,174,175]. In addition, inverting the hydroxyl group stereochemistry of hydroxyproline moiety abolishes all protein degradability of PROTACs [176,177]. Based on these facts, the authors rationalize the suitability of the tethering point for the photolabile group in a way that the tethering of the photolabile group would hinder VHL ligand binding to its VHL E3 ligase and can only be activated until irradiated (as shown in Figure 21B). The approach showcases an example of precise spatiotemporal control over photobiology. The author used carbonate tethering to substituted hydroxyl group of VHL with a diethylamino coumarin (DEACM) to form **ERRα PROTAC-2** (as shown in Figure 21C). In addition, the authors prepared a version of **ERRα PROTAC-2** that doesn’t have a DEACM photolabile group, **ERRα PROTAC-1**, as a control to assist in their biological study and ensure their photocaging PROTAC approach. Both PROTACs were designed to target an orphan nuclear hormone receptor (estrogen-related receptor α (ERRα)) [177], typically overexpressed in malignant cancers [178]. Using HPLC and mass spectrometry, the DEACM caging group cleaved from the **ERRα PROTAC 2** after 3 min of irradiation (λ ≤ 405 nm), and released the acidic functional groups with a pKa < 5 [179].

To understand the photocage PROTAC role of **ERRα PROTAC-2**, MCF-7 cells were treated with **ERRα PROTAC-1**, **ERRα PROTAC-2**, and DMSO in the absence/presence of UV radiation (λ = 365 nm, 180 s). After 8 h of incubation, the authors used Western blotting to measure the extent of ERRα cellular levels. As anticipated, **ERRα PROTAC-1** showed a significant ERRα protein reduction at the cellular level compared to the DMSO-based sample (used as control), which agrees with the previously reported literature [177]. Importantly, even a double concentration of **ERRα PROTAC-2** compared to **ERRα PROTAC-1** in the absence of UV light showed no change in ERRα protein cellular levels, confirming photocaging of photolabile group prevented the binding conformation towards E3 ligase, exemplifying an idealistic example of photocaged-PROTACs. To understand the mechanism, competitive assays in the presence of either the proteasome inhibitor (**MG132**) or the neddylation inhibitor (**MLN4924**) prevent the degradation of **ERRα PROTAC-2**, suggesting the proteasome- and E3 ligase-mediated degradation ability of **ERRα PROTAC-2**. Furthermore, incubation of MCF-7 cells with the coumarin caging group fragment released during photolysis showed no effects on ERRα levels, demonstrating that the observed degradation activity was highly mediated by the active, non-caged PROTAC generated via de-caging.

## 9. Conclusions

Classical medicinal approaches rely on orthosteric and allosteric inhibitors for direct target inhibition. Approaches such as (a) preferential protein subtype selectivity [180,181], or species-specific selectivity [182,183], (b) cyclization or rigidification in inhibitor design [184,185,186,187,188,189], (c) targeting heterodimeric proteins or receptors [12,190,191]), (d) identification of toxicophore to pharmacophore [192,193,194,195], (e) repurposing of drugs of clinical agents [196], (f) inverse screening [197,198]), and (g) exploration of natural phytochemicals [199,200,201] have gained interest in the recent decade. As these approaches focus on small molecule inhibitors (SMIs) and their related modes of action; therefore, often have issues that are typical with *occupancy-driven pharmacology* (shown in Figure 1A), such as (a) frequent occurrence of resistance after prolonged use and, (b) require a higher degree of potency to achieve full inhibition of protein of interest (POI). These issues led to drive *event-driven pharmacology,* as shown in Figure 1B, to design those molecules that can degrade the POI (known as protein degradation) and therefore diminish the related protein functions.

Selective targeting of the Estrogen receptor subtype constantly challenges the chemical biologist and medicinal chemist. However, cell-specific expression of Estrogen receptor subtypes (α, β, and γ) provides a selectivity handle to an extent (such as selective estrogen receptor modulators, also called SERMs). Classical approaches attempted were occupancy-based pharmacology, where the inhibitor (which could be heterocyclic, oligopeptide, or macrocyclic) shows average binding kinetics and requires continuous dosage administration. However, because of their continuous targeting nature, these approaches typically show their intrinsic flaws, where dose-dependent toxicity (off-target as well as on-target) and the emergence of resistance are the major ones. To minimize such interventions, selective estrogen receptor degraders (SERDs) were introduced in early 2000, which showed a significant improvement in ER-positive cancers and were popularized as the first line of drugs. Such success of SERDs validated the concept of estrogen receptor degradation. Due to the significant development of various types of protein degradation strategies, the researcher exploited the other estrogen ligands to functionalized them with additional activity. The construction of chemical chimeras shows three chemical entities, (a) one side of the structure contains the active molecules substructure or moiety that has an affinity to the estrogen receptor, (b) the remaining terminal of chimeras has the potential to bind the E3 ligase containing protein that recruits the ubiquitin units to the estrogen receptor, and (c) a chemical spacer or linkers that help to conjugate both the functionalities together. Various strategies were developed, including nonpeptide- and peptide-based versions of PROTACs and SNIPERs. However, these strategies undoubtedly bring high estrogen to subtype-selective but also have their flaws, such as on-target systemic toxicity; therefore, additional elements were incorporated into the PROTACs and SNIPERs: (a) Antibody conjugates SNIPERs and PROTACs, (b) Photocaged PROTACs. ADC approach is fruitful in minimizing the risk of molecular obesity of PROTACs/SNIPER; however, Photocaged PROTACs provide a handle to chemical biology to control the estrogen receptor spatiotemporal control.

However, there are certain limitations of peptide and non-peptide PROTACs and SNIPERs. For example, average lipophilicity, in vitro stability, and molecular obesity. Average lipophilicity and in vitro stability can be improved with drug delivery systems (antibody-drug conjugate, nanoencapsulation) [202,203], and synthetic procedures retrieved from combinatorial chemistry approaches [204,205,206]. However, to showcase an illustration of improved physicochemical properties, a joint venture of Arvinas and Pfizer developed a series of CRBN-based PROTACs (**ARV-110** and **ARV-471**) that showed high clinical effectiveness, as shown in Figure 22. **ARV-471** is orally bioavailable selectively ER-targeted PROTAC. In xenograft models, **ARV-471** showed more significant ER degradation and anticancer activity than fulvestrant. In addition, the phase-1 dose-escalation study demonstrated **ARV-471** as a tolerable single agent and showed anticancer benefit in ER+/HER2 breast cancer patients who were previously on hormonal therapy or cyclin-dependent kinase (CDK)4/6 inhibitor [207].

## Figures and Tables

**Figure 1 pharmaceutics-14-02523-f001:**
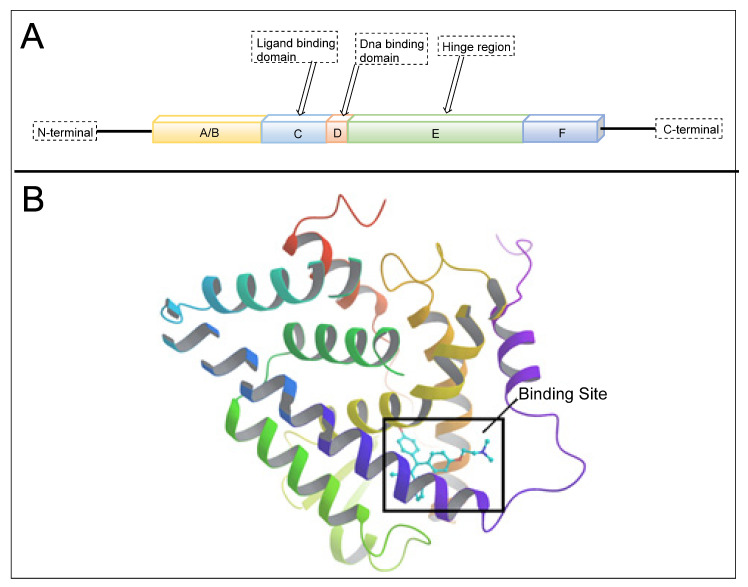
(**A**) Structure of estrogen receptor. (**B**) The binding site of ER-α.

**Figure 2 pharmaceutics-14-02523-f002:**
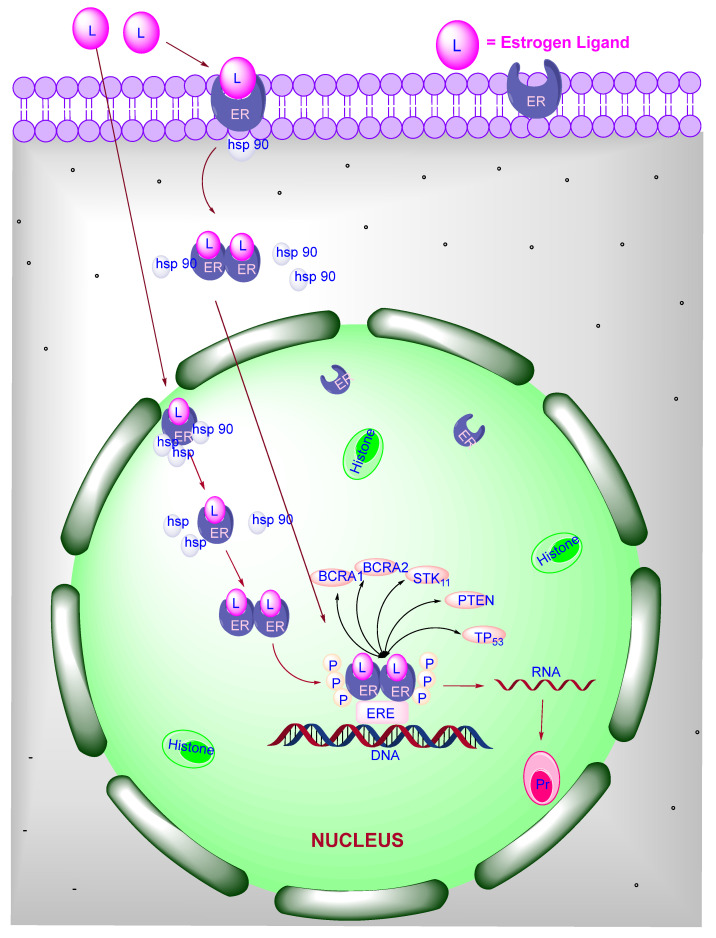
Mechanism of action: when a ligand binds with the ER, which is present on the cell membrane or phospholipid layer, the heat shock proteins associated with it are dissociate. Conformational changes take place and further proceeds to dimerization. Due to the receptor’s hydrophobicity, it enters the nucleus and binds with the ERE element on DNA. After that, it activates a variety of genes such as PTEN, STK 11 followed by the transcription process. Subsequently, translational process occurs. On the other hand, ligand (estradiol) directly enters into nucleus because of its hydrophobic nature. Next, the same process is also occurring here.

**Figure 3 pharmaceutics-14-02523-f003:**
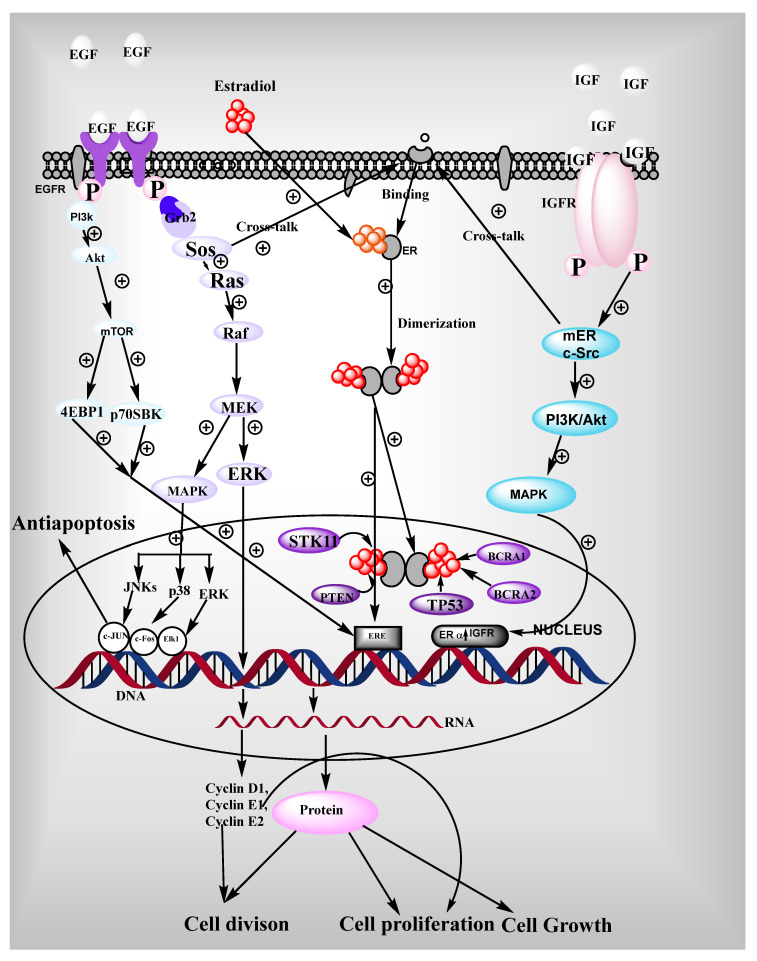
The binding of epidermal growth factor (EGF) with EGFR activates the receptor signaling due to phosphorylation. It stimulates various intracellular kinases signaling such as PI3K, Ras, MAPK, mTOR and induces cell division, proliferation, and growth. It interferes with other cellular signaling, such as IGFR, ER. It shows crosstalk with ER, which is present on the membrane, simultaneously cell signaling starts and the formation of proteins.

**Figure 4 pharmaceutics-14-02523-f004:**
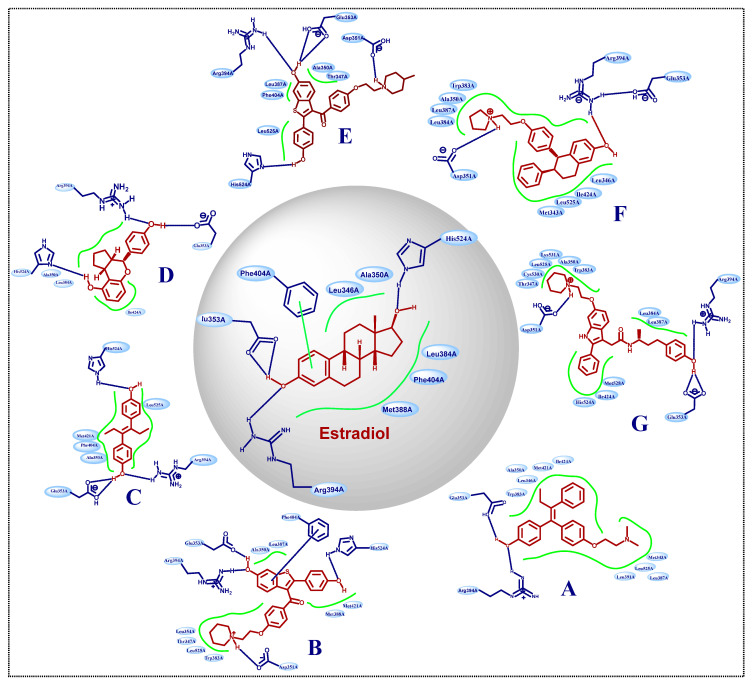
The interactions of ligands with ER-α receptor: Estradiol (PDB Id: 1GWR), (**A**) Tamoxifen, (**B**) Raloxifen, (**C**) Diethylstilbestrol, (**D**) Hexahydrocyclopenta[e]chromene, (**E**) Benzothiazine, (**F**) Naphthalen, and (**G**) Indole.

**Figure 5 pharmaceutics-14-02523-f005:**
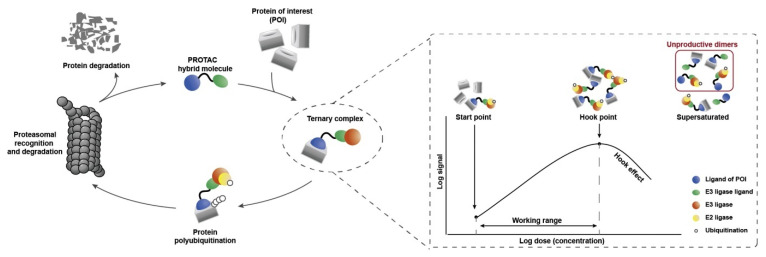
The generalized degradation of protein of interest (in this case, it would be ER-α) using proteasomal ubiquitination. Cooperatively plays an essential role in forming a ternary complex, which suffers from an indigenous issue that leads to the formation of a binary complex rather than the intended ternary complex (called the “Hooks effect”). It was reproduced with permission from Lin et al. [54]. Copyright 2020 Elsevier.

**Figure 6 pharmaceutics-14-02523-f006:**
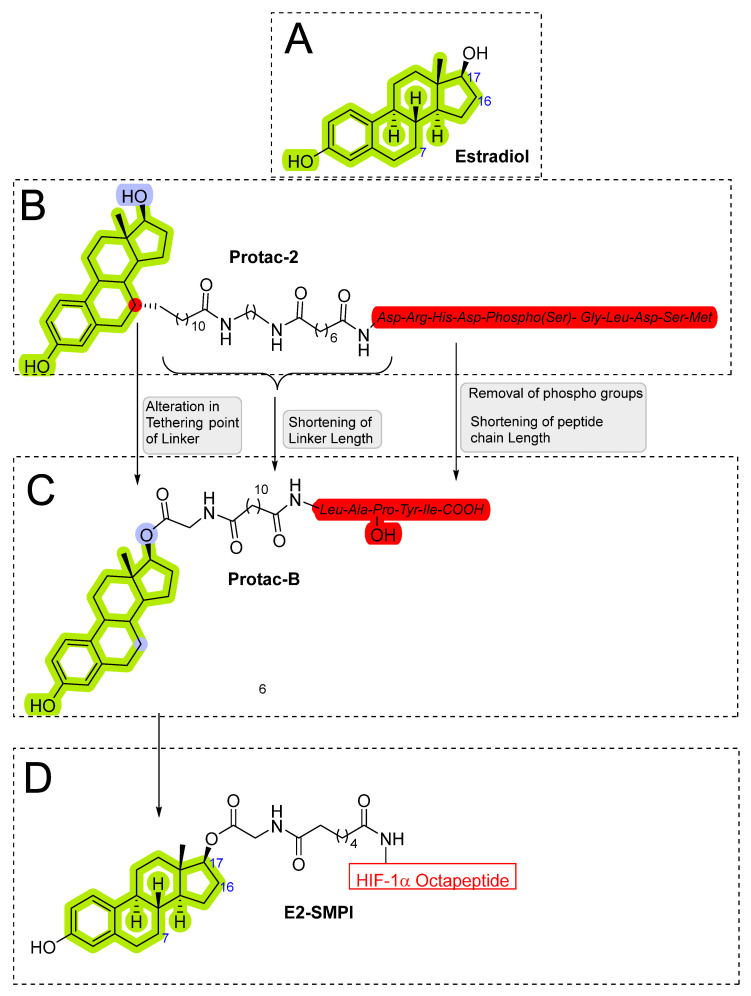
(**A**) Structure and numbering of Estradiol, a common ligand scaffold used for ER degradation. (**B**) Steroidal-based peptide PROTACs (**Protac-2**). (**C**) Chemical alteration yielded **Protac-B**. (**D**) Chemical structure of **E2-SMPI**.

**Figure 7 pharmaceutics-14-02523-f007:**
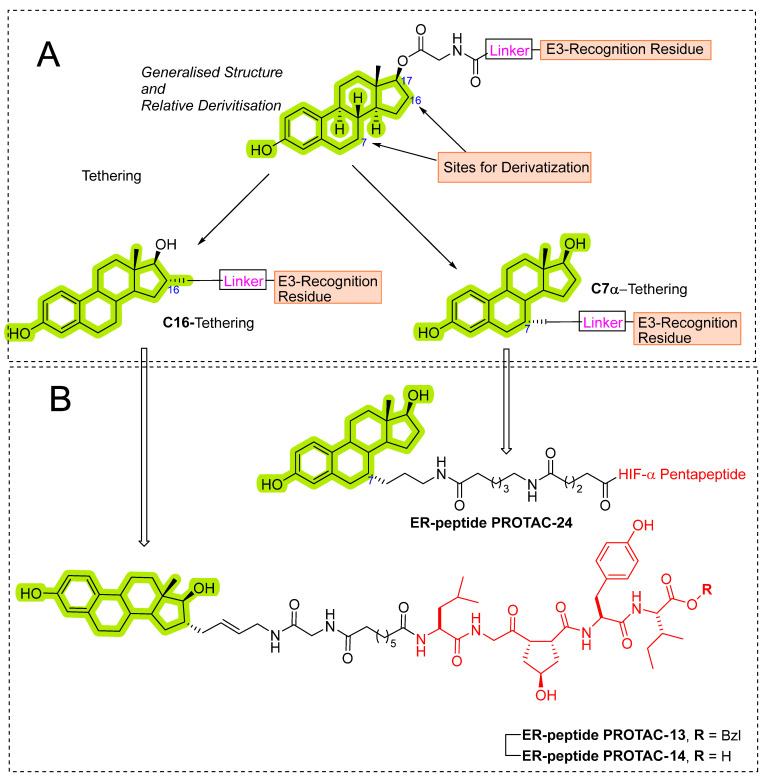
Development strategy for peptide-based ER-PROTACs. (**A**) Generalized structures and possible cases of derivatization. (**B**) Yielded peptide-PROTACs.

**Figure 8 pharmaceutics-14-02523-f008:**
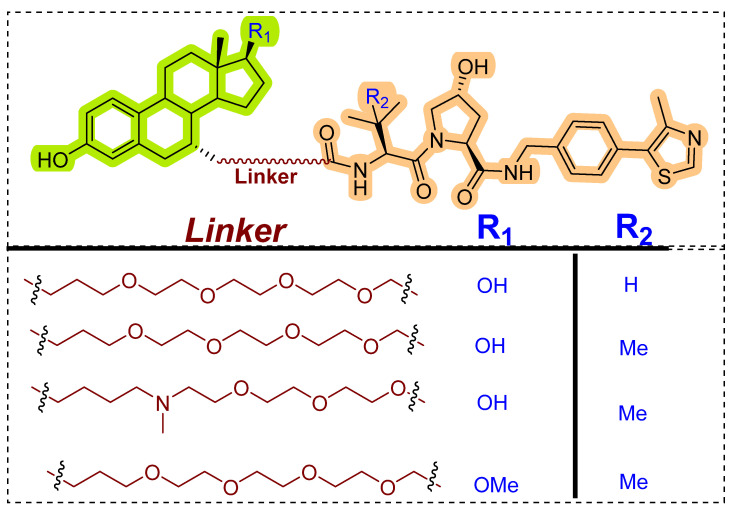
Generalized structure of Patented PROTACs by GSK.

**Figure 9 pharmaceutics-14-02523-f009:**
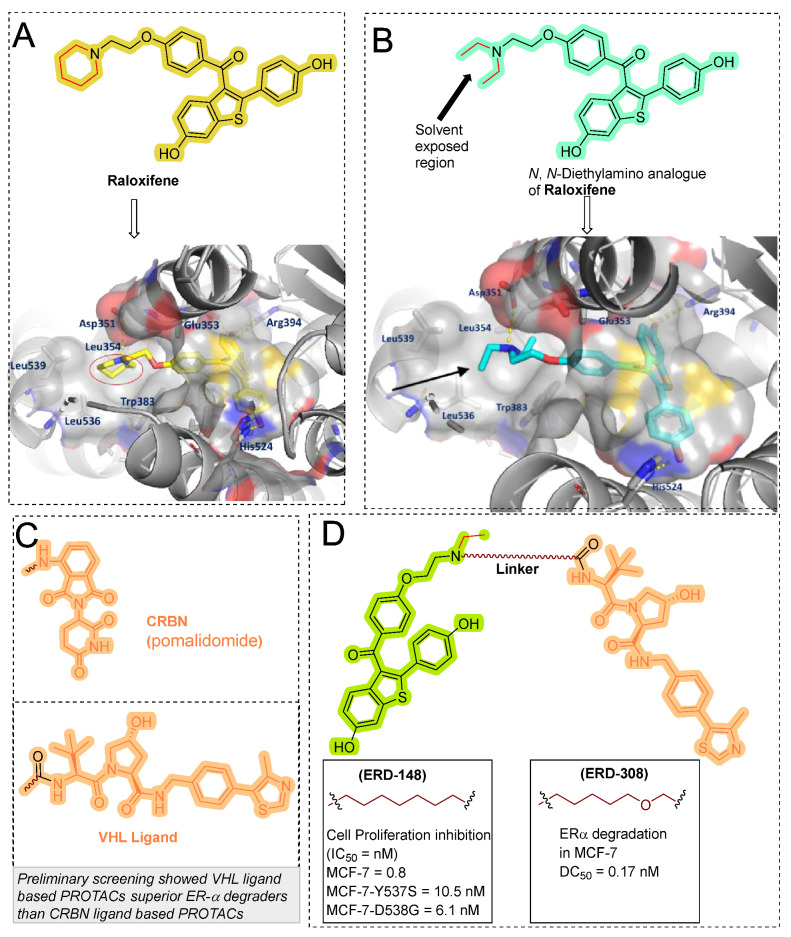
(**A**) Retrieving the conformational biding of raloxifene with ER-α (PDB 1ERR). (**B**) Predicted conformational binding of N, N-Diethylamino analog of Raloxifene with ER-α. (**C**) Molecular structure of CRBN and VHL ligand. (**D**) VHL-based PROTACs (**ERD-308**, **ERD-148**).

**Figure 10 pharmaceutics-14-02523-f010:**
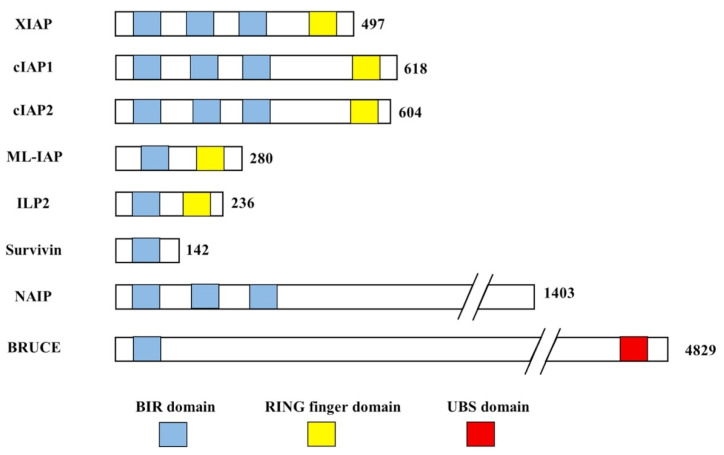
Available inhibitor of apoptosis protein (IAP) family proteins of humans. Reproduced with permission from Lin et al. [50] Copyright 2021 Elsevier.

**Figure 11 pharmaceutics-14-02523-f011:**
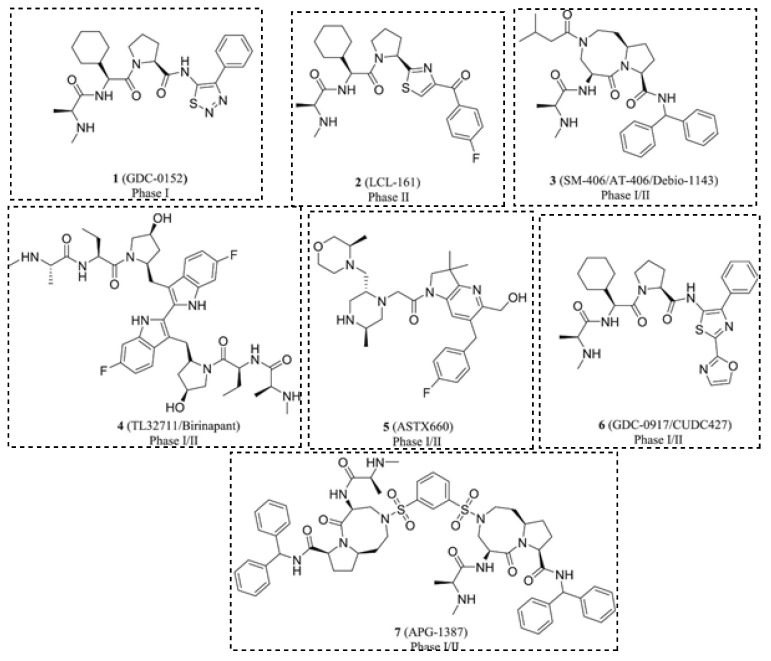
Reported IAP antagonists in clinical trials. Reproduced with permission from Lin et al. [50] Copyright 2021 Elsevier.

**Figure 12 pharmaceutics-14-02523-f012:**
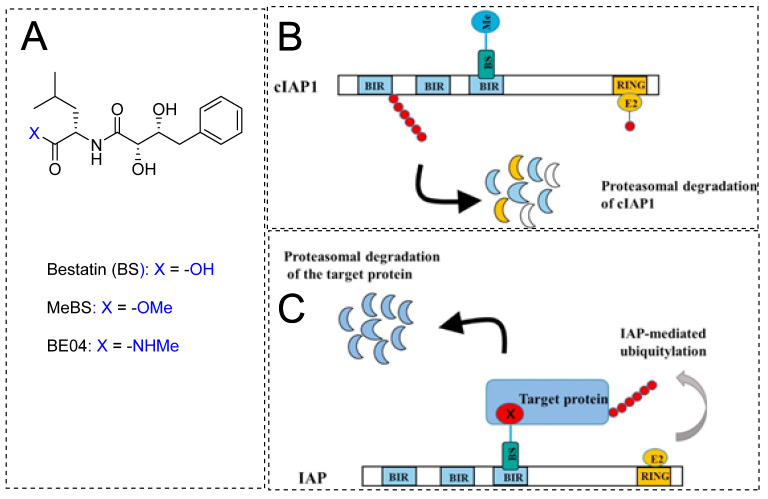
(**A**) Molecular structure of bestatin and its derivatives. (**B**) Reported degradation mechanism of cIAP. (**C**) Illustration of degradation associated with SNIPERs. (**B**,**C**) were reproduced with permission from Lin et al. [50] Copyright 2021 Elsevier.

**Figure 13 pharmaceutics-14-02523-f013:**
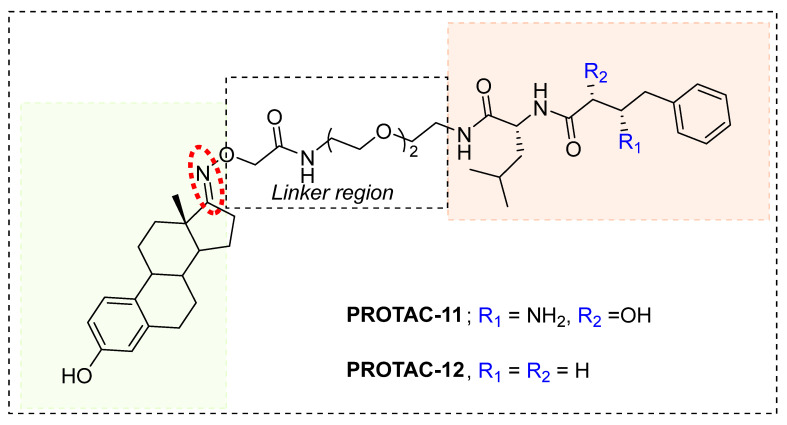
Molecular structures of **PROTAC-11** and **PROTAC-12**. Note: a typographical error in the linker length of **PROTAC-11** was found in Figure 11 from a review by Lin et al. [54] compared to the original structure reported by Itoh et al. (scheme 3) [114].

**Figure 14 pharmaceutics-14-02523-f014:**
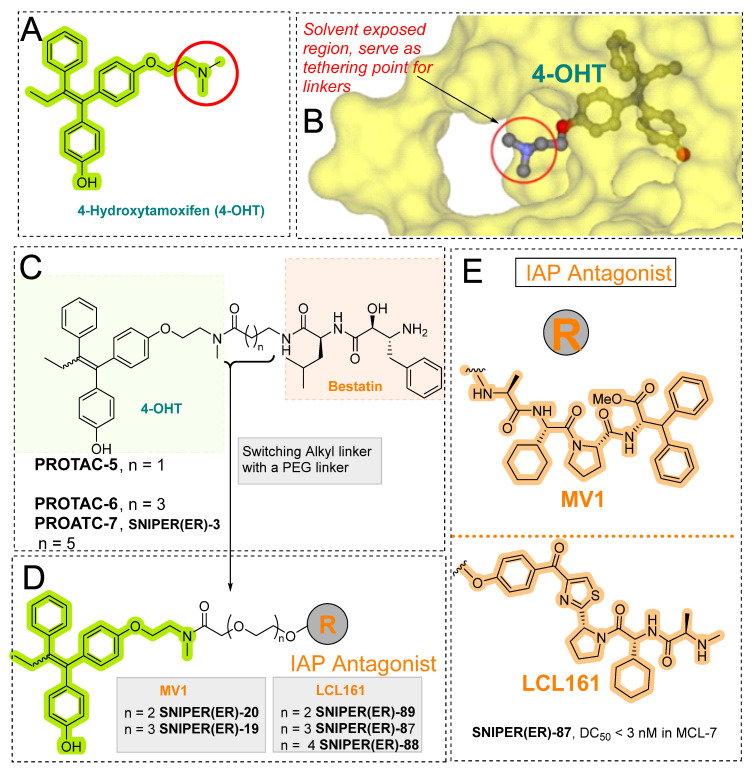
(**A**) Cocrystal structure (PDB id: 3ERT) of 4-hydroxytamoxifen (4-OHT) and ER-α, showing the solvent-exposed region of 4-OHT. (**B**) Molecular structure of 4-OHT. (**C**) Molecular structure of PROTAC-5, 6, and 7. (**D**) General structure of SNIPER(ER)s where linker of PROTAC-7 (**SNIPER(ER)-3**) was replaced with polyethylene glycol (PEG). (**E**) Molecular structures of IAP ligands.

**Figure 15 pharmaceutics-14-02523-f015:**
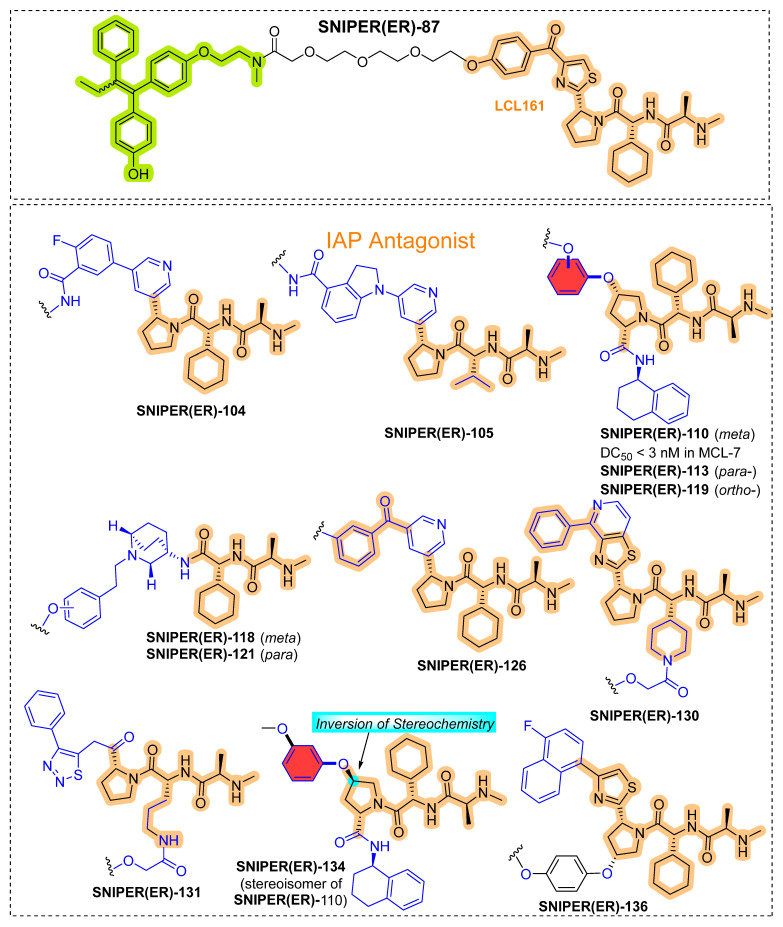
Various IAP structures studied to modify the **SNIPER(ER)-87**.

**Figure 16 pharmaceutics-14-02523-f016:**
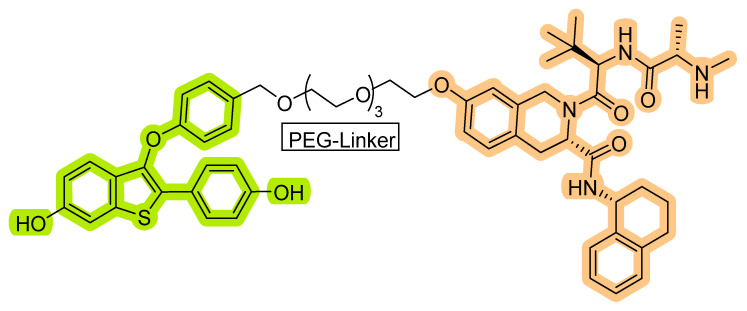
Patented SNIPER structure by GSK.

**Figure 17 pharmaceutics-14-02523-f017:**
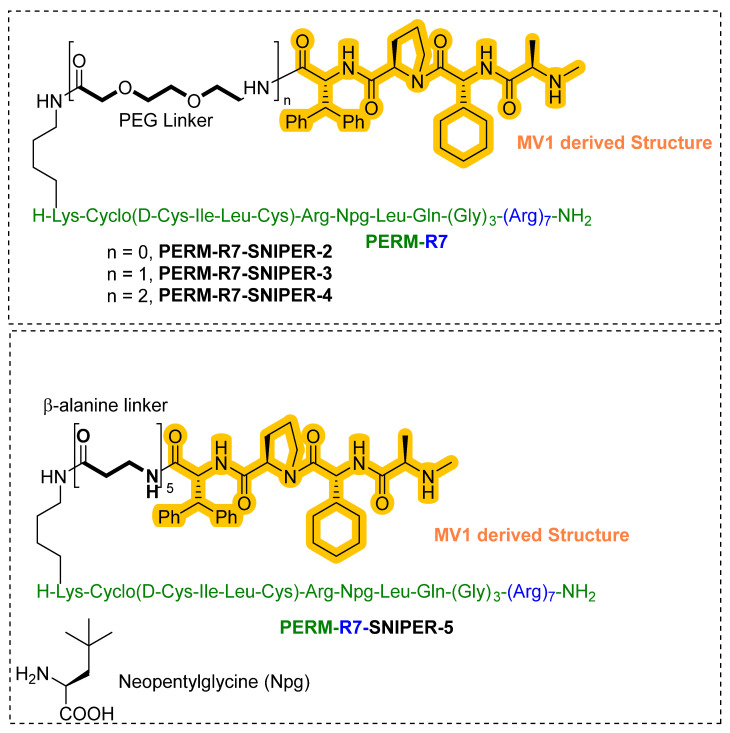
Molecular structure of PERM-R7 (marked in green color) based SNIPERs, which display a hepta-arginine fragment (marked in blue color) and MV1 derived structure (in orange color).

**Figure 18 pharmaceutics-14-02523-f018:**
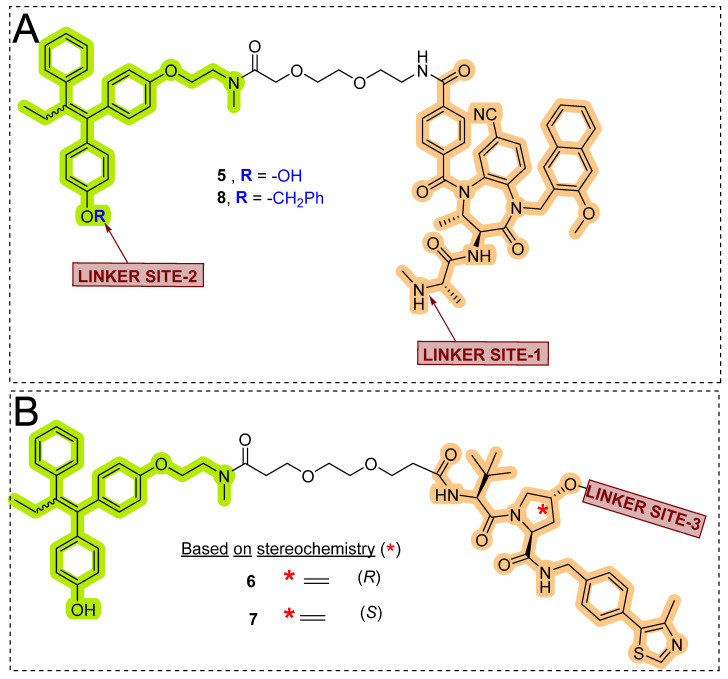
SNIPER and PROTACs used for ADC strategy. (**A**) Linker site-1 and 2 for ADC conjugation. (**B**) Linker site-3 for ADC conjugation.

**Figure 19 pharmaceutics-14-02523-f019:**
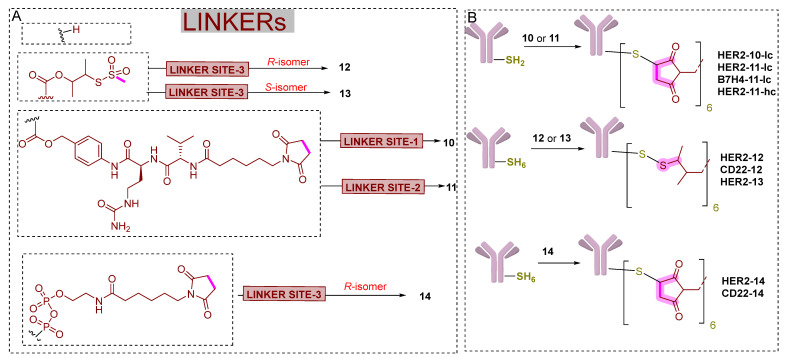
Linker types and conjugation strategies used for ADC designs. (**A**) Chemical structures of linkers. (**B**) Strategy illustration of linker ADC conjugation.

**Figure 20 pharmaceutics-14-02523-f020:**
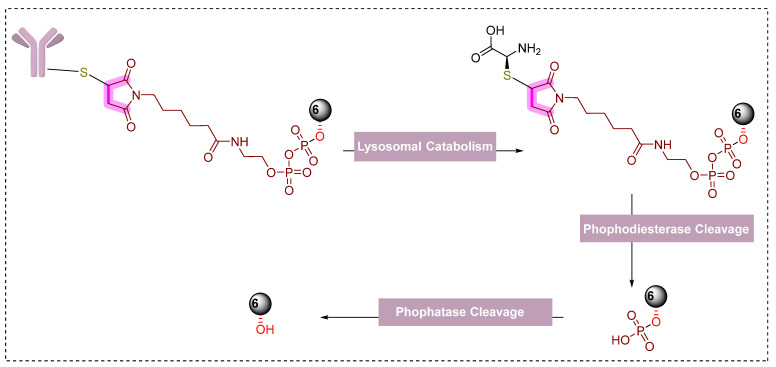
Improved ADC approach for delivery of **VHL-PROTAC-6**.

**Figure 21 pharmaceutics-14-02523-f021:**
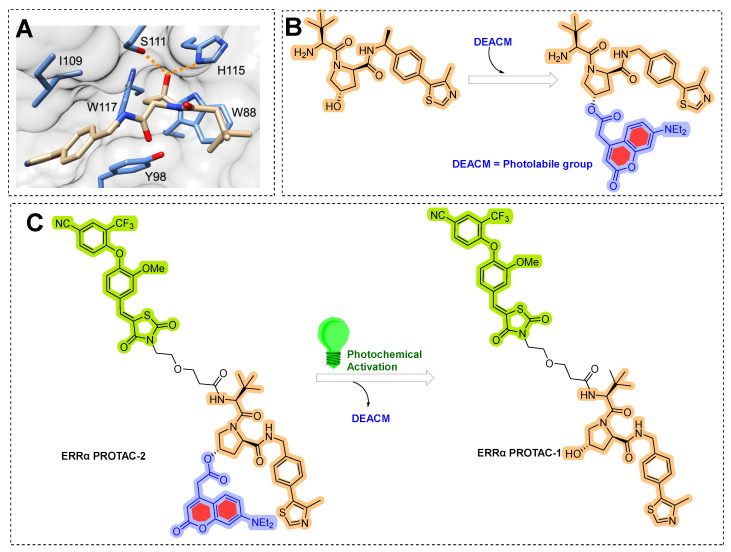
(**A**) Cocrystals VHL ligand binding to VHL E3 ligase, showing critical residue involvement in H-bonding interactions. (**B**) Chemical transformation of VHL ligand into coumarin-photocaged VHL ligand. (**C**) photochemical transformation of **ERRα PROTAC-2** to **ERRα PROTAC-1**.

**Figure 22 pharmaceutics-14-02523-f022:**
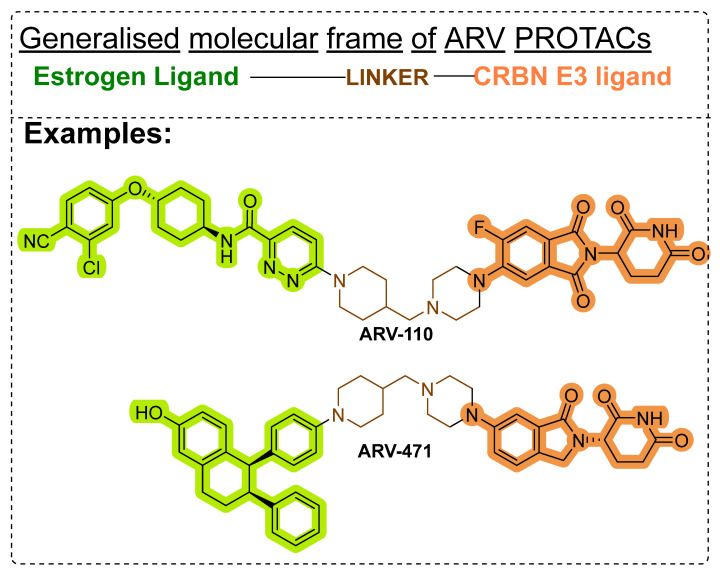
Molecular structure of **ARV-110** and **ARV-471**.

**Table 1 pharmaceutics-14-02523-t001:** Reported X-ray cocrystal structures of ER-α selective inhibitors.

PDB Entry	Estrogen Receptor Isoform	Inhibitor	Names of Inhibitors	Resolution (Å)	Reference
3DT3	Estrogen Receptor-α	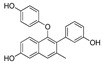	5-(4-hydroxyphenoxy)-6-(3-hydroxyphenyl)-7-methylnaphthalen-2-ol	2.40	[29]
2R6Y	Estrogen Receptor-α	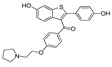	(6-hydroxy-2-(4-hydroxyphenyl)benzo[b]thiophen-3-yl)(4-(2-(pyrrolidin-1-yl)ethoxy)phenyl)methanone	2.00	[30]
2IOG	Estrogen Receptor-α	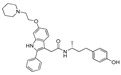	(R)-N-(4-(4-hydroxyphenyl)butan-2-yl)-2-(2-phenyl-6-(2-(piperidin-1-yl)ethoxy)-1H-indol-3-yl)acetamide	1.60	[31]
2IOK	Estrogen Receptor-α	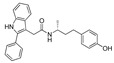	(R)-N-(4-(4-hydroxyphenyl)butan-2-yl)-2-(2-phenyl-1H-indol-3-yl)acetamide	2.40	[31]
2POG	Estrogen Receptor-α	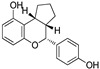	(3aS,4R,9bR)-4-(4-hydroxyphenyl)-1,2,3,3a,4,9b-hexahydrocyclopenta[c]chromen-9-ol	1.84	[32]
2Q70	Estrogen Receptor-α	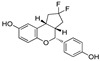	(3aS,4R,9bR)-2,2-difluoro-4-(4-hydroxyphenyl)-1,2,3,3a,4,9b-hexahydrocyclopenta[c]chromen-8-ol	1.95	[33]
2OUZ	Estrogen Receptor-α	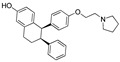	(5R,6S)-6-phenyl-5-(4-(2-(pyrrolidin-1-yl)ethoxy)phenyl)-5,6,7,8-tetrahydronaphthalen-2-ol	2.00	[34]
2AYR	Estrogen Receptor-α	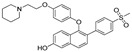	6-(4-(methylsulfonyl)phenyl)-5-(4-(2-(piperidin-1-yl)ethoxy)phenoxy)naphthalen-2-ol	1.90	[35]
1XQC	Estrogen Receptor-α	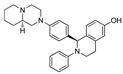	(S)-1-(4-((R)-octahydro-2H-pyrido[1,2-a]pyrazin-2-yl)phenyl)-2-phenyl-1,2,3,4-tetrahydroisoquinolin-6-ol	2.05	[36]
1XP6	Estrogen Receptor-α	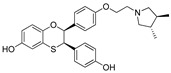	(2S,3R)-2-(4-(2-((3S,4S)-3,4-dimethylpyrrolidin-1-yl)ethoxy)phenyl)-3-(4-hydroxyphenyl)-2,3-dihydrobenzo[b][1,4]oxathiin-6-ol	1.70	[37]
1XPC	Estrogen Receptor-α	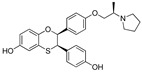	(2S,3R)-3-(4-hydroxyphenyl)-2-(4-((R)-2-(pyrrolidin-1-yl)propoxy)phenyl)-2,3-dihydrobenzo[b][1,4]oxathiin-6-ol	1.60	[37]
1R5K	Estrogen Receptor-α	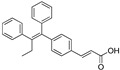	(E)-3-(4-((E)-1,2-diphenylbut-1-en-1-yl)phenyl)acrylic acid	2.70	[38]
1SJ0	Estrogen Receptor-α	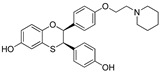	(2S,3R)-3-(4-hydroxyphenyl)-2-(4-(2-(piperidin-1-yl)ethoxy)phenyl)-2,3-dihydrobenzo[b][1,4]oxathiin-6-ol	1.90	[39]
1PCG	Estrogen Receptor-α	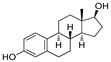	(8R,9S,13S,14S,17S)-13-methyl-7,8,9,11,12,13,14,15,16,17-decahydro-6H-cyclopenta[a]phenanthrene-3,17-diol	2.70	[40]
1UOM	Estrogen Receptor-α	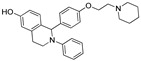	2-phenyl-1-(4-(2-(piperidin-1-yl)ethoxy)phenyl)-1,2,3,4-tetrahydroisoquinolin-6-ol	2.28	[41]
1GWQ	Estrogen Receptor-α	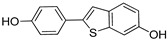	2-(4-hydroxyphenyl)benzo[b]thiophen-6-ol	2.45	[42]
1GWR	Estrogen Receptor-α	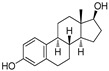	(8R,9S,13S,14S,17S)-13-methyl-7,8,9,11,12,13,14,15,16,17-decahydro-6H-cyclopenta[a]phenanthrene-3,17-diol	2.40	[42]
1QKU	Estrogen Receptor-α	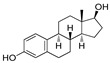	(8R,9S,13S,14S,17S)-13-methyl-7,8,9,11,12,13,14,15,16,17-decahydro-6H-cyclopenta[a]phenanthrene-3,17-diol	3.20	[43]
3ERD	Estrogen Receptor-α	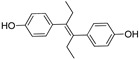	(E)-4,4′-(hex-3-ene-3,4-diyl)diphenol	2.03	[44]
3ERT	Estrogen Receptor-α	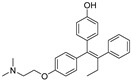	(E)-4-(1-(4-(2-(dimethylamino)ethoxy)phenyl)-2-phenylbut-1-en-1-yl)phenol	1.90	[44]
1A52	Estrogen Receptor-α	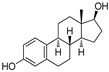	(8R,9S,13S,14S,17S)-13-methyl-7,8,9,11,12,13,14,15,16,17-decahydro-6H-cyclopenta[a]phenanthrene-3,17-diol	2.80	[45]
1ERE	Estrogen Receptor-α	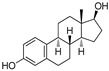	(8R,9S,13S,14S,17S)-13-methyl-7,8,9,11,12,13,14,15,16,17-decahydro-6H-cyclopenta[a]phenanthrene-3,17-diol	3.10	[46]
1ERR	Estrogen Receptor-α	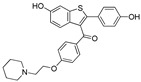	(6-hydroxy-2-(4-hydroxyphenyl)benzo[b]thiophen-3-yl)(4-(2-(piperidin-1-yl)ethoxy)phenyl)methanone	2.60	[46]

**Table 2 pharmaceutics-14-02523-t002:** In vitro binding affinities (IC_50_) of synthesized SNIPERs to ER-α and IAPs. The table was reproduced under the terms of the Creative Commons CC-BY license [124].

SNIPERs	ER-α DC_50_		IC_50_ (95% CI)				IAP Ligands	
	4 h	48 h	ER-α	cIAP1	cIAP2	XIAP	Developer	Reference
**SNIPER(ER)-87**	<3	83.0	110	450	960	700	Novartis	WO2008016893
**SNIPER(ER)-104**	>100	ND	61	>1000	>1000	>1000	Novartis	WO2012080260
**SNIPER(ER)-105**	<3	<3	69	5.3	7.2	55	Novartis	WO2012080271
**SNIPER(ER)-110**	<3	7.7	120	74	73	330	Abbott	[133]; WO2016169989
**SNIPER(ER)-113**	<3	13.3	150	85	99	810	Abbott	[133]; WO2016169989
**SNIPER(ER)-118**	>100	ND	230	140	900	>1000	AstraZeneca	[125]
**SNIPER(ER)-119**	4	15.7	200	80	48	700	Abbott	[133]; WO2016169989
**SNIPER(ER)-121**	>100	ND	ND	650	>1000	>1000	AstraZeneca	[125]
**SNIPER(ER)-126**	<3	3.7	83	68	200	490	Novartis	WO2008016893
**SNIPER(ER)-130**	36.9	ND	41	68	28	25	Genentech	[134]
**SNIPER(ER)-131**	33.8	ND	80	25	24	140	Genentech	[126]
**SNIPER(ER)-134**	>100	ND	47	550	360	>1000	Abbott	[133]
**SNIPER(ER)-136**	>100	ND	47	890	>1000	>1000	Genentech	[134]; WO2006069063

**Table 3 pharmaceutics-14-02523-t003:** PROTAC and SNIPER used for Antibody Drug Conjugates. Reproduced with permission from Dragovich et al. [143] Copyright 2020 Elsevier.

		MCF7-Neo/HER2 (ER-α)		MCF7 Parental (ER-α)	
Chemotypes	Ligase	DC_50_ (nM)	Dmax	DC_50_ (nM)	Dmax
**fulvestrant**	-	0.38 ± 0.07	84%	0.45 ± 0.01	87%
**SNIPER-5**	XIAP	1.6 ± 0.12	85%	0.68 ± 0.13	87%
**VHL-PROTAC-6**	VHL	4.9 ± 0.57	92%	2.7 ± 0.71	92%
**VHL-PROTAC-7**	VHL	7.9 ± 0.71	57%	5.2 ± 0.01	54%
**SNIPER-8**	XIAP	305 ± 103	76%	131 ± 2.1	75%
**endoxifen**	-	1.1 ± 0.37	48%	1.6 ± 0.01	49%

**Table 4 pharmaceutics-14-02523-t004:** Molecular binding characteristics of PROTAC and SNIPER-based Antibody Drug Conjugates. Reproduced with permission from Dragovich et al. [143] Copyright 2020 Elsevier.

Conjugate	DAR	Site	MCF7-Neo/HER2 (ER-α)		MCF7 Parental (ER-α)	
			DC_50_ (nM)	Dmax	DC_50_ (nM)	Dmax
**HER2-10-lc**	NA	LC-K149	ND	ND	ND	ND
**HER2-11-lc**	2.0	LC-K149	0.11 ± 0.001	81%	22 ± 3.4	71%
**B7H4-11-lc**	2.3	LC-K149	50 ± 23	40%	43 ± 16	62%
**HER2-12**	5.9	Multi	0.04 ± 0.007	99%	0.23 ± 0.07	95%
**CD22-12**	5.7	Multi	0.51 ± 0.094	90%	0.48 ± 0.16	93%
**HER2-13**	5.9	Multi	0.05 ± 0.016	87%	0.70 ± 0.068	51%
**HER2-14**	5.6	Multi	0.03 ± 0.002	94%	0.09 ± 0.013	95%
**CD22-14**	5.9	Multi	4.2 ± 0.078	70%	1.6 ± 0.035	91%
**HER2-mAb**	NA	NA	0.04	43%	>100	9%

## Data Availability

Not applicable.

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
