# Peer review of "Estrogen Receptor-α Targeting: PROTACs, SNIPERs, Peptide-PROTACs, Antibody Conjugated PROTACs and SNIPERs"

_pharmaceutics, 2022, doi:10.3390/pharmaceutics14112523_

Round 1

Reviewer 1 Report

Negi et al. submitted the article, "Estrogen Receptor-α Targeting: PROTACs, SNIPERs, Peptide-PROTACs, Antibody Conjugated PROTACs, and SNIPERs," is a focused review on degrading the estrogen receptor subtype selectivity through various protein degradation approaches. However, there are some estrogen reviews in this regard, but the author wrote a comprehensive review, which makes it different than others. 

The strength of the paper is in its writing flow which is quite systematically illustrated, making it different than other papers in the same field, where authors differentiate one protein degradation technique from the other. Secondly, the paper is written comprehensively, and the author also made some constructive points based on their observation, which are unique and informative.  

There are some minor comments. 

  1. I understand that It is not possible to cover each paper on ER-alpha protein degradation, but authors need to consider "ARV named PROTACs" (such as ARV-110 and ARV-471), which are clinical trials. However, as this paper is highly focused on PROTAC structures and structural activity relationship and, as there is very subtle information available on ARV structures, therefore that could be one of the reasons why the authors choose not to include ARV-based PROTACs. However, if the author adds a paragraph related to their clinical status, that will improve the scope of the paper. 

  1. PROTACs, SNIPERs etc., mentioned in the paper are based on event-driven pharmacology. However, some of the early career researcher (especially, chemical biologists/medicinal chemists) for whom this topic is new and not enough papers clearly defined the difference between event-driven pharmacology and occupancy-driven pharmacology; therefore, the authors needs to include a paragraph stating these differences, would improve the readability of the paper. 

  1. Although the paper's English is good, there are some sentences where the choice of words could have been better; please revise accordingly. 

The manuscript has all the merits to consider in the current journal if the authors agree to revise the points mentioned above. 

Author Response

Negi et al. submitted the article, "Estrogen Receptor-α Targeting: PROTACs, SNIPERs, Peptide-PROTACs, Antibody Conjugated PROTACs, and SNIPERs," is a focused review on degrading the estrogen receptor subtype selectivity through various protein degradation approaches. However, there are some estrogen reviews in this regard, but the author wrote a comprehensive review, which makes it different than others. The strength of the paper is in its writing flow which is quite systematically illustrated, making it different than other papers in the same field, where authors differentiate one protein degradation technique from the other. Secondly, the paper is written comprehensively, and the author also made some constructive points based on their observation, which are unique and informative. There are some minor comments. 1. I understand that It is not possible to cover each paper on ER-alpha protein degradation, but authors need to consider "ARV named PROTACs" (such as ARV-110 and ARV-471), which are clinical trials. However, as this paper is highly focused on PROTAC structures and structural activity relationship and, as there is very subtle information available on ARV structures, therefore that could be one of the reasons why the authors choose not to include ARV-based PROTACs. However, if the author adds a paragraph related to their clinical status, that will improve the scope of the paper. 2. PROTACs, SNIPERs etc., mentioned in the paper are based on event-driven pharmacology. However, some of the early career researcher (especially, chemical biologists/medicinal chemists) for whom this topic is new and not enough papers clearly defined the difference between event-driven pharmacology and occupancy-driven pharmacology; therefore, the authors needs to include a paragraph stating these differences, would improve the readability of the paper. 3. Although the paper's English is good, there are some sentences where the choice of words could have been better; please revise accordingly. The manuscript has all the merits to consider in the current journal if the authors agree to revise the points mentioned above.

Response to Reviewer:
We would like to thank the reviewer for taking the time and comment.

  1. we incorporated a dedicated section regarding ARV-110 and ARV-471 along with Figure 22 at the end of the conclusion section.
  2. We incorporated occupancy-driven pharmacology and event-driven pharmacology in the conclusion section.
  3. Also, we revised the manuscript text with a better choice of words

Reviewer 2 Report

The manuscript summarizes the compounds developed to degrade the estrogen receptor named PROTACs, SNIPERs and other combinations to be potentially used in breast cancer treatment. The review describes in detail the chemical structure of all the compounds and their functionality, with several figures that clarify the text. The quality of the figures is optimal.

Minor revisions:

11.    Some sentences deserve revision in the redaction and punctuation.

Line 190, verb seems to be missing: The X-ray cocrystal structures of ER-a in complex with estradiol (PDB id: 1GWR), as shown in Figure 4.

Line 267, the comma after "and" should be removed: The hydroxyproline-containing pentapeptide is not dependent on phosphorylation and, recognized by the VHL ubiquitin ligase.

Line 346, revise redaction of the sentence: Kinetics of ER degradation of ERD-308 studied in MCF-7 cells.

Line 365: Using quantitative reverse transcription-polymerase chain reaction analysis ERD-308 downregulates the of ER-regulated genes (pGR and GREB1) [84].

Line 616, see the underlined sentence: Therefore, these observed differences in those two cell lines might be resulted could be resulted from their different needs for IAPs.

22.    It could be informative that authors also add the names of the inhibitors listed in table 1.

33.    Authors could clarify the abbreviation "POI".

Author Response

We would like to thank the reviewer for taking the time and comment. The manuscript summarizes the compounds developed to degrade the estrogen receptor named PROTACs, SNIPERs and other combinations to be potentially used in breast cancer treatment. The review describes in detail the chemical structure of all the compounds and their functionality, with several figures that clarify the text. The quality of the figures is optimal.

Minor revisions: 11. Some sentences deserve revision in the redaction and punctuation.

Line 190, verb seems to be missing: The X-ray cocrystal structures of ER-a in complex with estradiol (PDB id: 1GWR), as shown in Figure 4.

Revised accordingly

Line 267, the comma after "and" should be removed: The hydroxyproline-containing pentapeptide is not dependent on phosphorylation and, recognized by the VHL ubiquitin ligase.

Revised accordingly

Line 346, revise redaction of the sentence: Kinetics of ER degradation of ERD-308 studied in MCF-7 cells.

Revised accordingly

Line 365: Using quantitative reverse transcription-polymerase chain reaction analysis ERD-308 downregulates the of ER-regulated genes (pGR and GREB1) [84].

Revised accordingly

Line 616, see the underlined sentence: Therefore, these observed differences in those two cell lines might be resulted could be resulted from their different needs for IAPs.

Revised accordingly

It could be informative that authors also add the names of the inhibitors listed in table 1.

Added the IUPAC names of the inhibitors in Table 1.

Authors could clarify the abbreviation "POI".

Revised accordingly in the abstract as well as in the main text.
